# Food Preservation in the Industrial Revolution Epoch: Innovative High Pressure Processing (HPP, HPT) for the 21st-Century Sustainable Society

**DOI:** 10.3390/foods13193028

**Published:** 2024-09-24

**Authors:** Agata Angelika Sojecka, Aleksandra Drozd-Rzoska, Sylwester J. Rzoska

**Affiliations:** 1Department of Marketing, University of Economics in Katowice, ul. 1 Maja 50, 40-257 Katowice, Poland; agata.angelika.sojecka@gmail.com; 2Institute of High Pressure Physics Polish Academy of Sciences, ul. Sokołowska 29/37, 01-142 Warsaw, Poland; ola.drozdrzoska@gmail.com

**Keywords:** high pressure preservation (HPP), high pressure and high temperature (HPT) sterilization, distortion-sensitive analysis for microorganism decay, barocaloric effect, sustainable society

## Abstract

The paper presents the ‘progressive review’ for high pressure preservation/processing (HPP) (cold pasteurization) of foods and the next-generation high-pressure and high temperature (HPHT, HPT) food sterilization technologies. It recalls the basics of HPP and HPT, showing their key features and advantages. It does not repeat detailed results regarding HPP and HPT implementations for specific foods, available in numerous excellent review papers. This report focuses on HPP and HPT-related issues that remain challenging and can hinder further progress. For HPP implementations, the reliable modeling of microorganisms’ number decay after different times of high pressure treatment or product storage is essential. This report indicates significant problems with model equations standard nonlinear fitting paradigm and introduces the distortion-sensitive routine enabling the ultimate validation. An innovative concept based on the barocaloric effect is proposed for the new generation of HPT technology. The required high temperature appears only for a strictly defined short time period controlled by the maximal pressure value. Results of the feasibility test using neopentyl glycol as the barocaloric medium are presented. Attention is also paid to feedback interactions between socioeconomic and technological issues in the ongoing Industrial Revolution epoch. It indicates economic constraints for HPP and HPT developments and emerging business possibilities. The discussion recalls the inherent feedback interactions between technological and socioeconomic innovations as the driving force for the Industrial Revolution epoch.

## 1. Introduction

Over 200 years ago, the Industrial Revolution epoch began [1]. The pattern of global civilization development has changed qualitatively due to the mass implementation of innovative technologies, the conceptual background of the Scientific Method, and feedback interactions with social changes [2,3]. The 1st Industrial Revolution, the Steam Age, extended even to the end of the 19th century [4]. Currently, the 5th Industrial Revolution is underway [5]. It is associated with the growing impact of artificial intelligence, the new generation of materials, and the sustainable, environment-friendly development pattern. For food products, it means production and processing with a minimized impact on the environment, low energy use, and loss-free storage and distribution. All these could significantly increase food quantity and quality, with lowered costs and, finally, consumer prices. Maintaining fresh product quality throughout the logistic chain is expected. Finally, the food should be of satisfactory quality and assortment and have pro-health features. It might seem an ‘ideal’ target. However, it can be reached during the fourth and fifth Industrial Revolutions times. This work focuses on an important element of this topic: innovative technologies for food preservation. The uniqueness of the Industrial Revolutions epoch shows the emergence of a new and surprisingly simple pattern for global population changes [6]:(1)$$P\left(t\right)=P_{0}exp\left(\frac{\;b^{\prime }\times t}{T_{C}-t}\right)$$
where the onset time t=0 is related to the year ~1700, i.e., the beginning of Industrial Revolution times, b’=const.

In the above relation, the prefactor is known, namely: $P_{0}=P\left(t=1700\right)$; the singular ‘Doomsday’ year $T_{C}$ can be determined from the analysis of the relaxation time τ(T) in the super-Maltus relation: $P\left(t\right)=P_{0}exp\left(t/\tau \left(t\right)\right)\;\Rightarrow \;\tau \left(t\right)=t/\left[ln\left(P\left(t\right)/P_{0}\right)\right]$, which yielded: $T_{C}=2216$ (year). Consequently, only the single adjustable parameter ($b^{\prime }$) in Equation (1) remains [6].

The first scaling equation for population changes was proposed by Thomas Malthus (1798, [7]):(2)$$P\left(t\right)=P_{0}exp\left(rt\right)$$
where r=const is the population growth rate coefficient.

Notable that in Equation (1), the rate coefficient is temperature-dependent, namely: $r={\left(T_{c}-t\right)}^{-1}$. Malthus linked population changes to available food amounts, for which the linear rising pattern was assumed [7]:(3)$$F\left(t\right)=A+Bt$$
where coefficients A,B=const.

Following Malthus, the ‘geometric’ population rise in Malthus Equation (2) is paralleled by a much weaker arithmetic increase in food resources (Equation (3)). In the 20th century, when long-term global population data were available, it became clear that the basic Malthus Equation (2) does not describe global population changes. Nevertheless, it has remained a significant conceptual reference [8,9,10,11,12,13,14,15,16].

A significant legacy of Malthus’s model constitutes the Malthusian Trap or Malthusian Catastrophe concept. It results from the simultaneous impacts of Equations (2) and (3), which Malthus commented as follows [7]: ‘*Population, when unchecked, increases in a geometrical ratio. Subsistence increases only in an arithmetical ratio*’, finally leading to times of ‘*vice and misery*’. Malthus advised to escape from the trap via population constraints or by an extra rise in subsistence, i.e., resources, including food. However, the simplistic concept of other countries’ robbery or conquest has often been ‘implemented’.

The Malthus model criticism started in the mid-19th century [8,9,10,11,12,13,14,15,16,17,18,19,20,21,22,23,24,25]. The Verhulst model was the first, and still important, as it explicitly included resources in the final scaling equation (1838–1847, [17]). It remains a significant inspiration in population-related studies also today [18,19].

In the second half of the 20th century, the cumulated impacts of the Industrial Revolutions epoch led to an unprecedented abundance and diversity of food despite the enormous growth in the global population following explicitly ‘super-Malthusian’ patterns, as Equation (1) for instance [6]. However, food shortages and even hunger have remained in large areas of our planet. They result from wars, socio-political disorganization, and global warming climate changes. Despite global-scale food abundance, global organizations cannot overcome local hunger disasters.

A question regarding pre-Malthusian times emerges. Very recently, the distortion-sensitive analysis via the super-Malthus equation [6]. Such an approach using Equation (2) with time-dependent growth rate $r\left(t\right)$ or alternatively relaxation time $\tau \left(t\right)=1/r\left(t\right)$ showed that from the year ~1200 to the Industrial Revolution’s onset: $\tau \left(t\right)=\tau =1050\;years=const$, with the Black Death worldwide pandemic distortion [6]. It means that from the late *Middle Ages* to the *Age of Reason and Enlightenment*, correlated with the Industrial Revolutions epoch onset, the simple Malthus behavior described by Equation (2) with r=const was a leading trend (!) [6]. A reliable discussion of food resources availability in this period is puzzling. However, one can draw some conclusions considering a country showing model features, such as: (i) unchanged borders; (ii) no invasions (i.e., devastating impacts) by neighbors; (iii) insignificant emigration and immigration; (iv) reliable and spacious historical documentation. In the mentioned period, England seems to meet these conditions. Available data shows that income per capita from the 13th century to the second half of the 17th century remained practically constant in England (!) [25,26,27,28]. It suggests a near-constant living standard and access to food resources for five pre-industrial centuries.

Near 1700, the Industrial Revolution epoch began. By 1800, per capita incomes in England doubled [25,26,27,28]. However, it was also a time of rapidly growing social inequalities, massive emigration constraining the population’s rise, and colonial conquest. The rapidly rising new industrial centers were inhabited by crowds of people, working 12–14 h daily and living in enormous poverty [1,8]. Such a world could motivate Malthus to forecast the catastrophic (Malthus) trap.

Food supply to numerous urban centers, created by boosting innovation-driven industries in this *Brave New World* [29], was essential. It meant a huge increase in food demands and complex logistics. For food, it also led to rising problems with taste, flavor, and health safety. In the second half of the 19th century, massive gastrointestinal diseases were pandemic in urban-industrial areas [30,31]. The quality of many products, especially taste, and flavour, became problematic.

The great writer Mikhail Bulgakov, in the epoch-hallmark book ‘*The Master and Margarita*’, euphemistically called it ‘*the second-class freshness food products*’ [32].

One of the responses to this challenge was the development of additives, such as sauces, that could mask the problematic features of foods. Some of them, such as ketchup, have survived, still enriching the cuisine [33].

The problem of food durability and health safety, inherently related to better use of food resources, has been known since prehistoric times. Many ‘historical innovations’ are still broadly used, enriching the cuisine nowadays. It includes drying, smoking, pickling, …. or a wide range of cheeses, for instance [34]. However, in the second half of the 19th century, there was a general call for a new generation of food preservation methods: maintaining basic ‘native’ features and simple and cheap in massive implementations.

A grand innovative breakthrough occurred due to Louis Pasteur (1863, [35]), who discovered that the rise of pathogenic microorganisms is the crucial factor responsible for foods and beverage quality deterioration and health hazards. He found that heating up to ~86 °C, for a few minutes can yield ~10^5^ (5-log) reductions of pathogen microorganisms. Later, the process was referred to as *Pasteurization* for Louis Pasteur’s recognition. Short-time heating to temperatures 130–150 °C leads to food’s ‘technological sterilization’ [36,37].

Louis Pasteur solved the grand problem plaguing humanity for millennia, namely how to protect water or beverages stored for a longer time so they remain safe and drinkable. The problem was often solved by adding alcohol as the preservative agent. In the Mediterranean countries, it was the diluted wine. In central and eastern Europe, vodka was added to water. Oceanic sailors preferred rum as a supplement. In some countries, beer was popular. The research of Louis Pasteur terminated this omnipresent, slightly drunken civilization era.

From the second half of the 19th century, preservation by using chemical additives was also developed. Nowadays, they are present in the majority of foods and beverages [38]. In the 19th century, the foundations of food cooling and freezing technology were also established. Today, this fundamental concept is essential for the myriad of refrigerators and freezers preserving food [36]. The mentioned innovative food preservation technologies were developed during the first and second stages of the Industrial Revolutions epoch. They solved grand problems regarding food safety and availability but also introduced significant threats that emerged in the last decades:Thermal pasteurization, ensuring microbiological safety, often significantly reduces products’ nutritional and bioactive properties [36,37].Chemical preservation additives appeared to be a factor supporting pandemics of obesity, some types of cancer, allergies, skin problems, and extreme intestinal problems [38,39,40,41,42,43,44].Cooling and freezing is an excellent preservation method [45,46]. However, it is still based on technological solutions that support the threat of Global Warming. The are aslo associated with huge energy consumption [47,48].

The solution to these problems is essential for the emerging Global Sustainable Society in the 5th Industrial Revolution epoch. It is possible due to the primary mechanism that drove the Industrial Revolution: A new generation of innovative food and beverage preservation & processing technologies. The basic innovation can be high pressure preservation/processing (HPP) technology for ‘cold pasteurization’, already market-validated. It also includes the next-step high pressure and high temperature (HPHT, HPT) technology for food sterilization [49,50,51,52,53,54,55,56,57,58,59,60,61,62,63,64,65,66,67,68].

This report presents the ‘progressive review’ of HPP and HPT technologies applied for foods and beverages. It does not repeat extensive tables showing the impact of HPP and HPT impacts on different pathogenic microorganisms for various food products. It does not repeat extensive discussions of the impact of pressure on selected components of food products. Such summaries, based on scans of literature data, can be found in numerous reviews. Selected reviews, published only in the last two years, are recalled in refs. [49,50,51,52,53,54,55,56,57,58,59,60,61,62,63,64,65,66,67,68].

First, the report recalls primary HPP and HPT issues to introduce and characterize the topic. Next, the report focuses on problems that have not been addressed so far. One can indicate those associated with the distortion-sensitive protocol for microorganism decay after high pressure processing or innovative solutions significant for overcoming problems hindering HPT technology. The proposed innovative solution is related to the barocaloric effect [47,48]. Finally, the meaning of HPP and HPT are discussed in frames of socio-economic changes during the Industrial Revolution and the Sustainable Civilization times.

## 2. Materials and Methods

Experimental results presented in the given report are related to microbiological tests of human milk contaminated with *Staphyllocossus aureus*, for which the preliminary presentations were published in refs. [69,70] by one of the authors et al. These reports also contain basic materials and method-related issues. High pressure treatments were carried out using a laboratory-scale HPP processor (made by UnipressEquipment, Poland) with the working volume of the pressure chamber V=1 L, generally operating up to P=1 L presented in Appendix A. In this report, the optimal treatment pressure was established in ref. [70], and tests were carried out for three selected temperatures. The results were analyzed via the novel-distortions-sensitive routine presented below. The target of the research was the reliable estimation of a parameterization of microorganism decay after different times of high pressure pulse lasting. Numerical filtering was carried out for the experimental data, enabling the differential analysis. It used the Savitzy–Golay-based routine implementation developed in refs. [71,72]. Regarding other results presented in the report, new conclusions and treatments are described in the subsequent sections.

## 3. Results and Discussion

### 3.1. HPP and HPT Revisted

#### 3.1.1. General HPP Features

This section presents essential facts regarding the history of HPP, its unique features crucial for the ongoing Sustainable Society epoch, and the essential characterizations and challenges for HPP processors’ technology. Significant supplementations are presented in the Appendix A.

Bert Holmes Hite carried out the first studies on the high pressure impact on foods at the end of the 19th century and reported them in the West Virginia Agricultural Experimental Station Bulletin in 1899 [73]. He presented the results of microbiological tests on milk subjected to pressures up to P=670 MPa for 10 min and showed a 5–6 log reduction in total counts. He also mentioned meat treated with pressure P=530 MPa for an hour. After three weeks of storage, only insignificant microbial growth was reported. In 1912, Percy W. Bridgeman reported egg albumin coagulation after compressing at P=590 MPa for 1 h [74]. This work launched an extraordinary research path: Bridgeman was honored with the Nobel Prize for high pressure studies in 1946 [75].

The works of Hite and Bridgeman [73,74] indicated the basic features of high pressures impacts on food products, namely the reduction in pathogen microorganisms and the denaturation of proteins. These studies explicitly indicated high pressure treatment as an alternative method for food preservation. However, industrial applications require a huge, multi-liter volume of high pressure chambers, and supporting facilities, operating for at least one hundred thousand compressing cycles without a primary service.

The first market food products after HPP (high pressure processing/preservation) appeared in Japan at the end of the 20th century [76,77]. They were related to HPP applications for strawberry, apple, and kiwi jams (Meidi-ya Corp., 1990) and grapefruit juice (Pokka Corp., 1991).

The global HPP food market is worth ~USD 7.5 billion, making it a leading next-generation innovative technology for food and beverage preservation [78]. This worldwide success is due to the unique benefits of HPP technology [49,50,51,52,53,54,55,56,57,58,59,60,61,62,63,64,65,66,67,68,69,70,79,80,81,82,83]:shelf-life extension from 2–3 days to even up to 180 days!high microbiological safetytaste, flavor, and appearance of a fresh productvitamin composition of the fresh productmaintaining bioactive propertieslimited or even no chemical preservativesactivation/deactivation of selected enzymessalt-free productsapplication to fluids and “soft, solid” foodapplication to packed food, reducing the risk of secondary contaminationenvironment-friendly technology, namely: (i) requirements for electric energy notably lesser than for thermal pasteurization; (ii) practical lack of waste during processingreduction of the number of expired products and, therefore also, disposal costsfor some consumers, important can be a ‘clean label’ and innovative technology

HPP technology is implemented via industrial-scale processors whose central point constitutes huge pressure chambers, with the volume available for products ranging from V=100 L even to V>1000 L. The HPP cycle begins with the initial phase, lasting 3–15 min, for placing the product in a container, which is next shifted to the chamber working volume. The chamber is then closed, filled with a pressure-transmitted liquid (usually water), and compressed to the planned process pressure. The latter is related to pressures from 300 MPa to 600 MPa. The main phase is keeping the product under the constant planned high pressure, lasting from 3 min to 10 min. Then, the final phase begins, namely decompressing, removing tens/hundreds of liters of water from the chamber, opening the chamber, and shifting the product away to the product feeder. This phase can last from ~several s for the sudden decompressing to ~5 min for the soft decompressing’. Note that significant electric energy consumption occurs only during the short compression phase.

Building industrial HPP processors constituted a technological challenge for decades. The basic problem was the large volume of the pressure chamber, the central part of the processor. A solution could be a scaling-up pressure chamber used in laboratory research, i.e., a monolithic cylinder made from a special steel with a hollow inside. However, for industrial-scale processors, the diameter of space available for products ranges from 20 cm to ~1 m and more. The length is from a few to a dozen meters. For such giant dimensions, the problem of minimizing compression-induced deformation, particularly significant in the central part of the chamber, is essential. It can be minimized by a design based on two cylinders tightly pressed into each other, with the outer one appropriately formed. Another solution, limiting compressing-related deformation after thousands of compressing cycles, is to wrap tightly the pressure chamber with many steel wires. Reducing parasitic deformation is also possible by limiting the maximum pressure, but such an HPP processor can be used only for selected types of products [83,84,85].

The pressure chamber has to be supplemented by a high pressures generating pump, yoke, process control to monitor temperature and pressure, a material handling system, and the control system. For safety reasons, the latter should be remote from the processor. Industrial applications above 100,000 compressing/decompressing cycles without the leading service are required [85].

Figure 1 shows the photo of the pilot-scale HPP processor with the volume chamber V=50 L, i.e., just below the industrial scale. It uses another protection against the mentioned deformation of high pressure chamber, namely wrapping the monolithic chamber with steel strips under appropriate tension. 

Such a concept also leads to 25% reduced total weight of the chamber compared to other solutions [85]. The design of the presented HPP processor ensures automatic loading and unloading of the product and the external control using the panel shown in the photo. The processor enables research and industrial-scale experiments. The HPP treatment of up to 1 ton of foods or beverages daily is possible, allowing for ‘market experiments’. See the Appendix A for supplementary views and comments.

The practical application of HPP technology involves isotropic compressing of a product placed in a flexible package for pressure transmission from the pressure medium, usually water, to the product. Such packaging, designed for a given product and made of a multi-layer polymers, should resist compressing and remain neutral for the product. During post-processing storage, it is necessary to avoid oxygen permeability. Regarding often chosen materials, one can indicate ethylene vinyl alcohol (EVOH) or polyvinyle alcohol (PVOH) copolymer [86]. However, polymer-based flexible packages raise the environmental problem. The optimal solution could be the usage of biodegradable polymers, but further studies are still needed. Considering the impact of pressure on foods and beverages, one should take into account the compressibility of water, which, at 25 °C, leads to ~14.7% volume decrease for P=600 MPa. It means ~8% decrease in the average distance between molecules or molecular assemblies. For the frequently used HPP implementations, pressure P=400 MPa can lead to ~5.5% decrease in the average distance between molecules. The dominant component of the vast majority of solid foods is water, but they also can contain proteins, fats, and micellar structures, which significantly increase the volume change upon compressing. Moreover, food products are essentially heterogeneous, i.e., composed of local mesoscale domains characterized by different compressibilities, always more prominent than for the background case of water [87].

#### 3.1.2. Food as a Soft Matter System

This section focuses on the links omitted in the monographic reports on HPP treated food systems, namely the so-called soft matter, which can fundamentally explain its strong sensitivity to compressing. Such an approach justifies the application of models and ‘tools’ developed in Soft Matter Science to the analysis of ‘foods under pressure’. It can be related to the description within the pressure-temperature plane curves associated with the denaturation of proteins or pasteurization decay of pathogenic microorganisms. Further, it supports identifying essential problems in the model description of the decay of microorganisms after the HPP treatment, finally offering new analytic tools to solve the problem.

In 1991, Pierre Gilles de Gennes established a novel category of materials named Soft Matter during his Nobel Prize lecture [88]. It includes liquid crystals, polymers, their blends, colloids, …, supercooled systems, supercritical systems, and even glasses [87]. The universal feature of such distinct systems is the dominance of mesoscale elements/assemblies, namely macromolecules or/and multimolecular assemblies. It leads to common scaling patterns for different physicochemical properties and great sensitivity to external perturbations, including pressure. Foods can be considered Complex Soft Matter [89], and animated microorganisms as Very Complex Soft Matter. The rising complexity increases the sensitivity to compressing, and microorganisms within food are particularly susceptible to such impact. The evidenced effects of pressure on microorganisms can be concluded as follows [50,53,56,57,79,80]:membrane disruptions, most often explained by local shear forces, but local differences in compressibility also can be importantproteins denaturationproteins deformation and local volume changes withindestruction of intracellular elements

This list shows that the mechanisms of pathogenic microorganisms destruction in food are more significant and complex for HPP than for classic thermal pasteurization, where the denaturation of proteins is the primary destructive mechanism [36,37,90].

Figure 2 shows ‘pasteurization’ curves in the P–T plane for three types of bacteria based on tests in the Agar matrix. They are related to the 5-log (10^−5^) decay after 10 min of compressing. Such curves strongly depend on the ‘matrix’ in which the microorganisms are located, i.e., they can slightly change for different food products. Celsius scale negative temperatures are notable. It is worth recalling that water can preserve liquidity under pressure for such temperatures despite the solidification at atmospheric pressure. Water exhibits the eutectic minimum at $\left(P\sim 210\;MPa,\;T\sim \;21.1\;{}^{\circ}C\right)$ [87]. In food systems, water is strongly locally trapped in different nano- and micro- constraints, which can lead to a ‘diffused’ eutectic minimum in P–T space. All these offer food storage under pressure at sub-zero temperatures without destructive water crystallization. It is worth emphasizing that maintaining the pressure inside the chamber after reaching the planned ‘stationary’ value does not require energy except for the current state monitoring.

The ‘pasteurization’ curve presented in Figure 2 correlates with the shape of the model curve describing protein denaturation, as shown in Figure 3, derived by Smeller and Herremans [91,92]. The relation describing the elliptic curve in Figure 3 was obtained using the extension of the classic thermodynamic Clausius-Clapeyron relation [91,92,93]:(4)$$\frac{dT_{s}}{dP}=T_{s}\frac{\Delta V}{L}=T_{s}\frac{\Delta V}{\Delta H}=\frac{\Delta V}{\Delta S}$$
where ΔV and ΔS denote the volume and entropy change associated with the transition between different states: in the given case, the denaturation; P stands for pressure and $T_{s}$ is for the characteristic temperature between two neighboring states; ΔH as for the enthalpy change for the given transformation/transition; it is related to the so-called latent heat L associated with the process.

Equation (4) was introduced to describe changes associated with discontinuous liquid-vapor and next liquid—solid crystal discontinuous transitions [91,92,93]. For the latter $T_{s}=T_{m}$, where $T_{m}$ is the melting temperature [87,93]. In such a case, Equation (4) is related to the state change between liquid and solid crystal states/phases.

Considering the simplest pressure-temperature dependence/expansion of ΔV and ΔS, one obtains [92]:(5)$$\frac{dT_{s}}{dP}=\frac{\Delta V}{\Delta S}=\frac{\Delta V_{0}+\Delta \beta \left|P-P_{0}\right|+\Delta \alpha \left|T-T_{0}\right|}{\Delta S_{0}+\Delta \alpha \left|P-P_{0}\right|+\left(\Delta C_{P}/2\right)\left|T-T_{0}\right|}$$
where $\left(P_{0},T_{0}\right)$ is the reference, onset pressure, and temperature; Δα is for the thermal expansion change; Δβ is for the compressibility change; and ${\Delta C}_{P}$.

It yields the elliptic functional dependence, schematically shown in Figure 3.

The denaturation curve shown in Figure 3 smoothly extends into the negative pressure domain, i.e., the isotropically stretched liquid [94,95]. A similar analysis leading to the elliptic curve in P–T space was also carried out for the melting temperature and the glass temperature [96,97].

#### 3.1.3. The Decay in the Number of Microorganisms after HPP Treatment and the Distortions-Sensitive Analysis

The HPP technology implemented for food and beverages dominantly focuses on microbial safety [49,50,51,52,53,54,55,56,57,58,59,60]. Compressing is associated with multiple destructive factors for microorganisms, as indicated above. The experimental evidence explicitly shows that the break of the cellular wall is crucial [51,54,60,61,67,70,79,80]. There is still no conclusive discussion regarding the physical origins of this phenomenon. For the authors, a possible appearance of local gelation or weak polymerization regions can be expected. The dominance of uniaxial elements in the wall, frustrated by lesser micro/nano elements, can also create local domains under supercritical conditions, inherently susceptible to even slight disturbances [93,98,99,100]. All these can support cellular wall breaks when compressing.

Studies for ‘designing’ optimal parameters for HPP treatment are primarily focused on microbiological safety. They are carried out in the following steps: (i) the contamination of the tested food by relevant microorganisms to a defined high level; (ii) action on the contaminated product via pressure pulses lasting $t_{1\;},\;t_{2,\;}$ …, for a given reference pressure and under selected constant temperature; (iii) post-processing number of microorganism estimations.

HPP is an innovative technology that requires establishing an optimal implementation scenario for each food product. The first goal of such optimization is to achieve the desired level of microorganism reduction, especially those hazardous to health. The name ‘cold pasteurization’ indicates the ability to reduce pathogenic microorganisms even by five decades, i.e., $\times 10^{-5}$ (5-log), by compressing even near the room temperature [49,50,51,52,53,54,55,56,57,58,59,60,61,62]. In parallel, the planned range of high pressures and the process temperature should be low enough to avoid disturbing essential product properties. Establishing such simultaneous conditions is particularly important for such a unique ‘food’ as human milk. It is the most important ‘model food’, with extraordinary sensitivity to any physical treatment. Notable that it is not only food but also medicine, and for newborns, an essential factor shaping the health for future life. It will be used to discuss the modeling features of HPP technology further in this chapter, basing the authors’ data on earlier studies in this field [81,82]. Human milk is a ‘product’ of extraordinary complexity and values, both nutritional and health-supporting. Human milk is the exclusive food for every person, at least for the first and often the second years of life. It shapes the health for the next decades of life. The results presented below are related to the author’s research on implementing HPP technology for human milk preservation.

Figure 4 presents data for human milk contaminated by *Staphylococcus aureus*, the essential and particularly hazardous type of bacteria in human milk [81,82]. The figure can also be representative of similar HPP tests of other kinds of food [49,50,51,52,53,54,55,56,57,58,59,60,61,62,63,64,65,66,67,68]. In such research, the model description of the microorganisms decay after HPP treatment is essential. A few model equations are applied to describe changes after different periods of high pressure impact [49,50,51,52,53,54,55,56,57,58,59,60,61,62,63,64,65,66,67,68]. The basic one recalls the basic Maltus Equation (2), i.e., it is related to the simple exponential function:(6)$$N\left(t\right)=N_{0}exp\left(-r\times t\right)\;\;\;\;\;\;\;\Rightarrow \;\;\;\;\;\;\;N\left(t\right)=N_{0}exp\left(-\frac{t}{\tau }\right)$$
(7)$$lnN\left(t\right)=lnN_{0}-r\times t=lnN_{0}-\frac{1}{\tau }\times t\;\;\;\;\;\;\;\;$$
where $N_{0}$ is the initial reference population, t is related to the time for which the maximal, stationary pressure value is applied, r is for the population change rate coefficient (decay or rise), and τ=1/r is the relaxation time.

Most often, for results presented via the ‘Maltus scale’ ${log}_{10}N\left(t\right)$ vs. t; notable that ${log}_{10}N\left(t\right)=lnN\left(t\right)/ln10$. The right part of Equation (6) is expressed via the relaxation time, τ=1/r which is related to $t=N\left(t=\tau \right)/N_{0}=e^{-1}$ (e=2.718…) decay of microorganisms. 50% decay is related to $\tau_{50\%}=\tau \times ln2$ value. 

The semi-log plot can validate the description following Equations (5) and (6)), if a linear behavior appears. It can be called the Malthusian behavior. Generally, the relaxation time is not used in microbiological tests, although it offers a useful interpretation of results.

Figure 4 can be considered an illustration of standard results for HPP studies reported so far [49,50,51,52,53,54,55,56,57,58,59,60,61,62,63,64,65,66,67,68,69]. Their characteristic features: (a)There is a limited number of data: usually, tests are carried out for 4–6 time periods of tested high pressure pulses. Nevertheless,, a nonlinear pattern of changes is commonly observed, indicating the non-Malthusian behavior, i.e., beyond Equation (6).(b)Non-Malthusian behavior is analyzed via the empowered exponential function (see below), with a heuristic recalling the Weibull model. However, the graphical presentation of fitting results, essential for the validation, is hardly presented.(c)Fitting results are typically given only in tables, with questionable estimations of parameter errors and fitting quality parameters, in the opinion of the authors.

Figure 4 shows the results of fitting via the empowered exponential/Weibull-type equation with realistic error estimations, as discussed below. They are qualitatively larger than most of the reported [49,50,51,52,53,54,55,56,57,58,59,60,61,62,63,64,65,66,67,68,69].

(d)Generally, the number of experimental data points in nonlinear fitting should be a decade larger than the number of fitted parameters. For the empowered exponential/Weibull equation, there are three adjustable parameters. Hence, the formally correct analysis requires at least 20 data points. Such results are (very) hardly reported.(e)The meaning of parameters derived via nonlinear fittings is usually not discussed.

The above issues indicate common problems with nonlinear fitting analysis for the decay of microorganisms in HPP-related studies. They show problems associated with the basics of experiments, which should take into account essential features and constraints of fitting routines. The discussion addressing the above problems is presented below. It also introduces innovative distortion-sensitive and derivative-based analytic validating tools. Finally, the recently proposed double exponential model [101] approach is discussed.

When experimental data follows the non-linear pattern, visible in Figure 4, the output dependence for the statistical Weibull model related to the dynamics of complex systems is often recalled [49,50,51,52,53,54,55,56,57,58,59,60,61,62,63,64,65,66,67,68]. It can be presented as follows [6]:(8)$$N\left(t\right)=\frac{\beta }{\alpha }{\left(\frac{t}{\alpha }\right)}^{\beta -1}exp\left[-{\left(\frac{t}{\alpha }\right)}^{\beta }\right]\Rightarrow$$
(9)$$\Rightarrow N\left(t\right)=N_{0}exp\left[-{\left(\frac{t}{\tau }\right)}^{\beta }\right]\Rightarrow$$
(10)$$\Rightarrow {log}_{10}\frac{N\left(t\right)}{N_{0}}=-b\times t^{\beta }$$
where $N_{0}$ is for the initial reference value, $,\;\alpha \;\beta \;and\;b=\frac{1}{\tau }=const$ are model parameters.

Equation (8) presents the basic Weibull dependence, where the first power-type term is important only for the short-time decay. Only the long-time empowered exponential time scale is significant for the phenomena discussed in the given report. Following ref. [6], in Equation (9), the substitution for the relaxation time α→τ is introduced. Equation (10) is related to Equation (9) in the semi-log representation, generally used for discussing microorganism decay in microbiological tests [49,50,51,52,53,54,55,56,57,58,59,60,61,62,70]. The empowered exponential parameterization is most often implemented via the non-linear fitting. Such results are shown by curves interpolating experimental data in Figure 4. Parameters for portraying experimental data via Equations (9) or (10) can also be determined using a simple linear regression fit, namely using the following data transformation [6]:(11)$$\begin{array}{ll} {-log}_{10}\left(N\left(t\right)/N_{0}\right)= & b\times t^{\beta }\;\;\Rightarrow \;\;{log}_{10}\left(N_{0}/N\left(t\right)\right)=b\times t^{\beta }\;\;\;\Rightarrow \;\;\;\\ & \Rightarrow \;\;\;{log}_{10}\left[{log}_{10}\left(N_{0}/N\left(t\right)\right)\right]={log}_{10}b+\beta \times {log}_{10}t \end{array}$$

Figure 5 shows the results of the linearized derivative analysis via Equation (11), using data from Figure 4. Visible is the arbitrariness, reflecting the large experimental errors for results of the nonlinear fitting given in Figure 4. It illustrates the general problem for the standard discussion regarding the modeling of microorganism decay in HPP technology, which can be noted in reference reports [49,50,51,52,53,54,55,56,57,58,59,60,61,69,70,79,80], namely:(a)Usually, results of fitting via Equation (8) or Equation (9) are not presented in plots but only given in a table.(b)Experimental data are only linked by straight lines to facilitate the view.(c)Usually, post-HPP decay experimental data are related to 4–5 tested lengths of high-pressure pulses related to data points in the plot.(d)The errors of fitted parameters presented in the tables are usually significantly lower than those given in Figure 4.

The last issue is worth commenting. The fitting results presented in Figure 4 are related to the ORIGIN software, and the nonlinear fitting using the Marquardt protocol focused on minimizing. $\chi^{2}$ factor, describing the relative standard deviation for subsequent experimental data points concerning the tested scaling function, which shows the fitting reliability. It is worth stressing that a reliable fitting requires ~10x larger number of data points than the number of fitted parameters. Notable that to show the fitting quality, the correlation factor $R^{2}$ is most often given [49,50,51,52,53,54,55,56,57,58,59,60,61]. It is suggested that $R^{2}\to 1$ should indicate the fair correlation between the experimental data and model scaling relation. Such values are given in Figure 4, but the value $R^{2}\approx 1$ poorly correlates with fitting quality. The $\chi^{2}$ parameter focused analysis is a better metric, commonly used in complex systems physics [71,72,87]. For the authors, the above brief discussion shows that the debate on post-HPP decay of microorganisms requires revisiting and even changing the analytic routine paradigm.

In the analysis of post-HPP decay via the empowered exponential Equations (8) or (9), it is often assumed that the reference prefactor $N_{0}$ is perfectly known. In the authors’ opinion, this assumption can be questioned, and the uncertainty regarding the initial factor $N_{0}$ is similar to N(t) decay values. 

Recently, in ref. [6], the new way of population data analysis using Equations (8) and (9) yielding optimal values of τ and β parameters, with reliable error estimations and avoiding the knowledge of $N_{0}$, was proposed [6]. Namely:
(12)$$lnN\left(t\right)=lnN_{0}-{\left(\frac{t}{\tau }\right)}^{\beta }\;\;\;\Rightarrow \;\;\;\frac{dlnN\left(t\right)}{dt}=\frac{\beta }{\tau^{\beta }}{\left(\frac{t}{\tau }\right)}^{\beta -1}$$

The latter leads to:
(13)$${log}_{10}\left[\frac{dlnN\left(t\right)}{dt}\right]={log}_{10}G_{N}\left(t,N\right)={log}_{10}\left(\frac{\beta }{\tau^{\beta }}\right)+\left(-1\right){log}_{10}\left(\frac{t}{\tau }\right)=\;\;\;=\left[{log}_{10}\left(\frac{\beta }{\tau^{\beta }}\right)+\left(\beta -1\right)-{log}_{10}\tau \right]+\left(\beta -1\right){log}_{10}t=A+B{log}_{10}t$$where the final parameters A,B=const if the empowered exponential behavior can be used for portraying N(t) empirical data. 

Notable that the coefficient $G_{N}=dlnN\left(t\right)/dt=\left(dN/N\right)/dt$ can be considered as the analytic counterpart of the relative ‘per capita’ or ‘per element’ population growth factor. The linear behavior for the plot ${log}_{10}G_{N}\left(t\right)$ vs. ${log}_{10}t$ validates the portrayal via Equation (8). The application of linear regression in the plot based on Equations (12) and (13) for estimating parameters *A* and *B* determines optimal β,τ values. The substitution of these values to Equation (8) yields an unequivocal estimation of $N_{0}$ via the single parameter fitting. Figure 6 presents the results of such analysis for post-HPP decay in human milk contaminated with *S. aureus*, after stationary compressing at P=400 MPa, in subsequent periods. For optimizing experimental data and obtaining their smooth set, the preliminary numerical filtering based on the Savitzky–Golay principle, developed by the authors in ref. [71,72], has been applied. Fitting results via empowered exponential Equation (8) are shown in Figure 6, with related parameters. The realistic small values of the estimated errors and superior tracing of experimental data by model-related curves are notable. 

Notably, the empowered exponential Equation (8) also recalls the Kohlraush–Williams–Watts (KWW) model, used in complex systems physics, soft matter physics, and biophysics. For such co-notations, the exponent β>1 is for the compressed relaxation, and β<1 is for the stretched exponential processes [6]. Following the KWW approach, one can estimate $\tau_{50\%}=\tau^{\beta }\times ln2$ [6].

Figure 6 presents the results of HPP studies in human milk contaminated with *Staphylococcus aureus* after different time lengths of high-pressure pulses ${\Delta t}_{i}$ related to the value P=400 MPa, for temperatures given in the plot. The figure shows the complete set of data, which is preliminarly presented in ref. [81], with supplementary numerical smoothing. Note the output data quality, which is well above the standards used in similar studies carried out so far. 

Recently, the biphasic model has been developed to analyze post-HPP decays related to the different stationary compressing times and describe dynamics in some specific complex systems. The decay analysis is carried out via the following relation [49,101]:
(14)$${log}_{10}N\left(t\right)={log}_{10}N_{0}+{log}_{10}\left[fexp\left(-k_{1}t\right)+\left(1-f\right)exp\left(-k_{2}t\right)\right]$$where *f* is the fraction of the pressure-sensitive population, is the fraction of the resistant-to-pressure population, $k_{1}$ and $k_{2}$ inactivation rates for these two populations, and t is the treatment. In the authors’ opinion, Equation (14) adequacy for the given task is limited. First, four adjustable parameters significantly increase the fitting uncertainty. Second, during the HPP treatment for the given time period, pressure-sensitive microorganisms (related to $k_{1}$) are destroyed, and they are not detected in the post-processing analysis. Only decreasing fractions of pressure-insensitive (to given process conditions) are registered for subsequent HPP. All these seem to be not consistent with the HPP process implementations.

The standard thermal preservation technology for human milk is called Holder pasteurization or ‘soft pasteurization’. It is associated with processing at 65 °C for 30 min. Destructions in nutritional, bioactive, and immunological properties caused by the standard pasteurization at T~86 °C, are non-acceptable for human milk. The temperature T=50 °C, is the highest temperature considered in HPP tests on human milk, where the non-desired thermal pasteurization impacts can be avoided [81,82].

Compressing under adiabatic conditions leads to a homogeneous rise in temperature. However, in real HPP processors, this extra thermal energy is dominantly dissipated to the surrounding, via the pressure chamber. It is associated with compressing, but a notable time is always required when using a pressure pump, facilitating thermal energy dissipation. The final temperature rise after reaching the planned stationary pressure value is relatively low and quickly dissipates. A parallel process, but associated with temperature lowering, occurs when decompressing. However, quasi-adiabatic conditions appear for a sudden decompressing, lasting <1 s, and leading to a relatively strong cooling. See comments and the graphical illustration in ref. [81] for studies using the semi-lab scale HPP processor shown in the Appendix A.

The applications of HPP technology for market-related foods and beverages are focused on microbiological safety. It is worth noting that for destructing spores and viruses, pressures in the range of 0.8–1.4 GPa are required [49,51,54,56,66], making the process impractical for industrial implementations. Such extreme compressing can also be destructive for many food products and reduce the durability of HPP processors. A remarkable illustration may be the impact of compressing on human milk, the essential model food for the whole life. Taking into account only the microbiological safety 600 MPa might seem to be optimal. However, it leads to the substantial destruction of human milk’s essential properties [81,82]. Hence, the maximal processing pressure has to be adjusted to preserve human milk features and acceptable microbiological safety. It turned out that optimal is using two properly correlated high pressure pulses, with a maximum pressure value of ~350 MPa. The appropriate shape of the high pressure pulses is also important. This scheme, proposed by one of the co-authors of this work (SJR), was called HPP+.

Results regarding the microbiological safety of such an approach are presented in refs. [81,82]. Figure 7 illustrates the impact of the standard thermal Holder pasteurization and HPP+ treatment on primary human milk constituents compared to the native milk. It is notable that HPP+ treatment can even enhance the activity of some significant constituents. All these show pro-health ‘food engineering’ possibilities associated with HPP technology, opening gates for new products. However, the individual preparation of HPP implementation for a given product can be advised.

As noted above, the unique set of characteristics for products subjected to HPP technology, highly beneficial to both the consumer and the consumer, makes its limitations more evident. One can indicate:
(i)HPP technology does not lead to food sterilization.(ii)HPP rarely deactivates undesirable enzymes and often even increases their activity. However, for the last several years, research on high pressure and high temperature (HPHT, HPT) technology has been carried out, which allows for overcoming the above problems.

The next section refers to HPT technology as a significant extension of HPP addressing these problems. Nevertheless, there are still questions regarding the practice of the HPT method, which hinders its broad implementation. An innovative solution to overcome this problem is presented.

### 3.2. High Pressure and High Temperature Sterilization Technology (HPT, HPHT)

#### 3.2.1. HPT Basics

This section presents the basics related to the thermal sterilization of foods and its pressure-assisted counterpart, HPT (HPHT) technology. It also comments on issues related to adiabatic heating, which is inherently coupled to compressing. Technological problems associated with industrial applications are indicated.

Appropriately ‘designed’ chemical additives can lead to technological sterilization, i.e., maintaining microbiological safety, taste, and flavor, for a year or longer. This method is crucial to the abundance of goods on store shelves.

However, long-term consumption of such prepared foods reduces essential nutritional properties and has led to dangerous health consequences [39,40,41,42]. Thermal sterilization ensures microbiological safety. It can even deactivate some undesired enzymes, but to an even greater extent than thermal pasteurization, it reduces bioactivity and nutritional value [90]. There are two basic schemes for thermal pasteurization [90]: (i) wet pasteurization at a temperature of 121–125 °C for ~15 min in the presence of saturated vapor, and (ii) short-tome (~1 s) treatment with a temperature above 150 °C. It is suggested that increasing compression up to ~1.4 GPa in HPP technology can destroy bacteria, spores, and viruses, thus leading to sterilization [56,66,67,84,102,103]. However, it means the essential rise of HPP processor costs, limiting its durability and increasing the costs of service inspections. Extreme pressures can also be destructive to essential food properties. In this context, the maximum pressure P=600 MPa of standard HPP industrial processors available today seems to be a favorable compromise. It is also worth paying attention to the impact of P~1 GPa pressures on polymer-based packaging required in HPP technology. Experimental evidence shows that pressures slightly above 1 GPa applied to some basic polymers can lead to pressure-created polymerization without any supporting activator [87]. Hence, the risk related to extreme pressures’ impact on materials used for packing food products should be considered.

For several years, the high-pressure & high-temperature (HPHT, HPT) treatment has been indicated as an innovative sterilization technology, overcoming some limitations of the HPP [84]. It combines compressing up to P=400–600 MPa, used in standard HPP processors, with temperatures of 80–130 °C. Numerous research reports indicate the benefits of HPT technology [84,102,103,104]:Spores inactivation—efficiently inactivating spores to enable shelf-stable products.Nutrient preservation—preserving nutrition and flavor better than standard heat methods.Reducing newly formed contaminants such as heterocyclic aromatic amines, acrylamide, and n-nitroso- compounds.Efficiency: less operating energy than for classic thermal pasteurization can be neededReaching technological sterilization under ‘softer’ conditions than for the classic thermal pasteurization, regarding the treatment temperature and the time of its application.The reduction of enzyme activity

The latter issue is worth commenting on. For HPP, enzymes are often intact, or their activity increases. It can lead to undesired changes in color, texture, and flavor, influencing the stability of the chilled product. The evidence shows that after HPT treatment, the enzymes can mostly be rendered inactive, resulting in a stable color and texture during shelf life. Avoiding some enzymatic components is advised for some people because they can pose a health risk.

Testing the simultaneous impact of pressure and temperature, as required in HPT technology, is relatively simple for laboratory-scale experiments. However, implementing HP and HT conditions on large-scale industrial, or even pilot-scale, processors remains a challenging problem. Only recently, a practical solution using standard HPP processors has been proposed [105]:(i)The pre-heating product up to ca. 90 °C is placed in a thermally isolated container;(ii)The container is shifted to the standard HPP pressure chamber;(iii)The adiabatic compression up to P=600 MPa further increases the temperature of the product up to ca. 120–130 °C. Such conditions are preserved for ca. 5 min;(iv)Decompressing and returning to the ambient conditions;(v)Product removal from the chamber

Adiabatic temperature changes (i.e., under thermal isolation from the surroundings) are worth commenting. It is a universal phenomenon leading to homogeneous heating when compressing and parallel cooling when decompressing. The following relation describes the process [81,93]:(15)$$\frac{T_{final}}{T_{initial}}={\left(\frac{P_{final}}{P_{initial}}-1\right)}^{\left(\gamma -1\right)/\gamma }$$
where $\gamma =c_{P}/c_{V}$ denotes the ratio between the specific heat under constant pressure and temperature.

In reports regarding HPP, the issue is most often concluded by the statement that water compressing is coupled to $\left(3K–4K\right)/100\;MPa$ temperature rise [49,50,51,52,53,54,55,56,57,58,59,60,61,62,63,64,65,66,67,68]. The presence of complex structures, which are often essential for foods, can increase the temperature rise. On decompressing, a corresponding adiabatic reduction in temperature takes place. However, it is only a rough estimation. The above relation allows for the precise estimation of adiabatic heating/cooling, although it requires the knowledge of $c_{P}$ and $c_{V}$ pressure dependencies.

The HPT industrial solutions cope with significant implementation problems. For the authors, the most interesting is the mentioned recent solution [105]. However, one should indicate that the product is under a relatively high temperature, at least ~90 °C, for a relatively long time, when taking into account the product loading, compressing, decompressing, and unloading periods. Below, we present an innovative solution to solve some puzzling problems, minimizing the high temperature action time. 

The concept employs the so-called barocaloric effect. It is associated with inherent features of discontinuous phase transitions between less ordered and more ordered phases. The classic example is freezing water into solid ice on cooling and solid ice melting into liquid water on heating. Essential properties associated with passing the phase transition are described by the Clausius–Clapeyron Equation (4). It indicates basic process characterization, namely: (a) entropy change ΔS for describing ordering (‘symmetries’) change when passing the transitions, (b) latent heat L for the heat released from the system or absorbed by the system from the environment during the phase transition. These properties are equally important for temperature- and pressure-related passing the discontinuous phase transition, which is also shown by Equation (4). This equation also indicates the importance of the pressure dependence of the melting temperature, which for the vast majority of systems increases with compressing [98].

#### 3.2.2. Barocaloric Effect-Based Innovative Solution for HPT Sterilization

This section presents an innovative solution for HPT technology applications, exploring the so-called barocaloric effect for heating the product. The appearance of the high temperature is controlled solely by planned high pressure values, and the product can be ‘cold’ at the unloading stage. The results of the preliminary feasibility test are presented in the subsequent section.

In 2000, Strässle, Furrer, and Müller [106] pointed out that compression leading to the passage of a discontinuous phase transition can turn a system into a more ordered state, which can be considered a kind of ‘cool storage’. During decompression, the stored ‘cool’ is released. It is associated with the fact that the system has to absorb the energy for the environment to transform into a less ordered (‘more chaotic’) state. It leads to a temperature decrease in the surrounding space if it is -limited and thermally isolated (adiabatic). The barocaloric effect is considered a potentially superior concept for the new generation of refrigeration devices [47,48]. ‘Cool’ storage occurs after a relatively short compressing time. The ‘storage of ‘cool’ and its release require almost no power (electric energy) supply. No potential harmful environmental impacts. Commonly used classical refrigeration technology is based on the nearly continuous circulation of special (volatile) fluids subjected to decompression during the circulated flow. It requires an almost continuous power supply. Applied fluids are harmful to the atmosphere and ‘support’ Global Warming. However, the cooling ‘efficiency’ of the materials used in this process is colossal, reaching entropy change $\Delta S\approx 300\;JK^{-1}{kg}^{-1}$ (process metric) [47,48], For two decades since ref. [106] appearance, studies of the barocaloric effect in various materials had resulted in ΔS values ~50× lower, which precluded a meaningful discussion on applications.

In 2019, a colossal barocaloric effect was discovered for the plastic crystal–solid crystal discontinuous phase transition in neopentylglycol (NPG). At a pressure of ~450 MPa, even $\Delta S\approx 400\;JK^{-1}{kg}^{-1}$ was reached [47,48]. NPG is cheap, readily available, non-toxic, and environmentally safe material. It is used even in the food and pharmaceutical industries. In the following years, $\Delta S\sim 800\;JK^{-1}{kg}^{-1}$ for the barocaloric effects when passing the discontinuous phase transitions in soft matter systems on compressing, for instance, in macromolecular alkanes, were noted [107,108,109]. It is associated with the unusual richness of complex symmetries (ordering) arrangements occurring in soft matter systems during a strongly discontinuous transition.

The authors propose using the barocaloric effect in soft matter systems, such as NPG, for the new generation of HPT technology. Heat is released to the surrounding space when the discontinuous phase transition is passed on compressing. Under space-limited adiabatic conditions, it has to raise the surrounding temperature by ΔT. This value depends on ΔS and the amount of the material. On decompressing from the more ordered phase, the surrounding is cooled by ~ΔT. Only the second stage (cooling) is assumed to be used for hypothetical future refrigerating devices recalled above.

The innovative extension of the HPT technology can be concluded as follows:The standard HPP processors can be used for the innovative HPT-barocaloric concept, but the surface of the pressure chamber interior should be covered by a layer of thermo-isolated material, for instance Teflon. It ensures near-adiabatic conditions within the chamber.The processed food product is placed in standard containers used for HPP technology, supplemented (‘mixed’) with the barocaloric effect exploring elements, for instance, as shown in Figure 6 (see comment below).The compression/decompression, as for the standard HPP processing, starts.However, when passing the ‘designed’ pressure, the interior of the chamber and the product are heated due to the barocaloric heat effect freed on compressing.Additional heating is associated with general adiabatic heating. It is effective due to adiabatic conditions.“Designed” thermal parameters are maintained for the Δt time when the planned high pressure value acts.Decompression: the product is cooled due to the barocaloric effect on decompressing and the adiabatic cooling on decompressing.A relatively ‘cool’ product is shifted from the chamber to the product transporter. It can be immediately placed in a chilling room if necessary.

Additionally, a cylindrical container where the processed product is placed can play an active role. It can consist of two pipes of slightly different diameters. The space between them can be filled with a barocaloric material. Cutting out the outer pipe and wrapping the entire element in this place with an elastic band will allow for effective pressure transmission.

For the described innovative method of temperature control, adiabaticity, i.e., thermal insulation of the entire HPP processor pressure chamber volume from the environment and a properly selected material exhibiting the barocaloric effect, are essential. 

Studies in recent years have shown that using a material from the soft matter family is optimal. An example can be neopentyl glycol (NPG), where a discontinuous phase transition occurs from the ODIC type plastic crystal phase (mechanically resembling plasticine) to the solid crystalline phase (mechanically resembling a ‘stone’). This compressibility difference significantly contributes to entropy change ΔS when passing the discontinuous transition. The next contribution results from the great number of elements of symmetry in the adjacent phase.

The purpose of using such material for the ‘barocaloric HPT’ is that the soft material should be modeled appropriately and prepared, i.e., to prepare for the release of thermal energy for the planned maximum pressure $P_{max}$ of the discussed HPT process. Initially, the pressure dependencies $T_{f}\left(P\right)$ (freezing temperature) and $T_{m}\left(P\right)$ (melting temperature) should be determined, and then the entropy changes associated with them. Such data constitute a prerequisite necessary for determining the reference data so that during compression after reaching the pressure $P_{max}$ the transition to the stationary phase lasting Δ*t* occurs, combined with a slightly earlier release of thermal energy. Here, another helpful feature of soft matter systems is significant changes in the temperatures of phase transitions with even a tiny endogenous supplementation. Its appropriate selection also allows for control of the hysteresis between $T_{f}$ and $T_{m}$. Examples of such solutions are given in the patent application concerning the ‘barocaloric HPT’ innovation by the authors of this work. The aforementioned preliminary studies related to soft matter ‘barocaloric material’ were possible in the authors’ laboratory: X-PressMatter for pressurized soft matter and model foods IHPP PAS [110].

For the proposed innovation, the high temperature appears only when the planned high pressure value is reached. The initial temperature of the processed product may be typical of those used in a standard HPP application, e.g., 20–45 °C. After leaving the pressure chamber, the product will have a similar temperature, facilitating all further activities. The proposed solution provides great flexibility for ‘designing’ high temperature and pressure processing values. There is no significantly higher energy consumption compared to standard HPP technology.

Figure 8 shows a proposed ‘barocaloric element’, in which temperature rises on passing the process onset pressure. This results from the action of ‘latent’ heat associated with a discontinuous phase transition from a more disordered phase (e.g., liquid) to a more ordered phase (e.g., solid crystal). 

In the preliminary feasibility tests, the authors used neopentyl glycol (NPG), where the discontinuity between the ‘orientationally chaotic, liquid-like’ plastic crystal phase and the solid crystal phase occurs. When decompressing and shifting from the solid crystal phase to the plastic crystalline phase, energy must be taken from the environment, substantially reducing the temperature.

NPG is placed in a tube made of a highly thermally conductive material, such as copper or its alloys. A flexible element at its end can be thermally welded. It enables pressure transmission from the surrounding area, which transmits liquid to the element’s interior. The flexible tube ensures immediate transfer of pressure inwards. This method of pressure transmission was developed by the authors of this work during dielectric tests of liquids, as shown in ref. [98]. Such ‘elements’ can be dispersed between processed food products and/or linked to the walls of the container where the food product is located. The proposed ‘element’ is simple, cheap, and easy to repair or replace. The barocaloric element can also supplement the construction of the large tube in which processed products are placed. Notably, due to temperatures emerging during the treatment, pure water cannot be used as a pressure-transmitting liquid. Higher temperatures appear only within the pressure chamber’s interior, covered by a temperature-isolated material such as Teflon. Hence, it has no parasitic impact on the chamber body and other elements of the pressure system.

The barocaloric effect is considered the most promising concept for the next generation of ‘cool storage’ for refrigerators and air conditioners. Nevertheless, there are still no pilot implementations of such technology due to the so-called hysteresis and the problem of transporting ‘heat’/‘cool’ from the chamber to the chamber.

These factors are irrelevant to the proposed innovative HPT barocalorics-based solution. This fact and technological simplicity may cause it to be the first barocaloric effect implementation in practice.

#### 3.2.3. Preliminary Feasibility Test for the ‘HPT-Barocaloric’ Innovative Solutions

The above innovative application of the barocaloric effect requires experimental validation, which is the target of this section. An important element of the innovation mentioned above is the transformation of the interior of the pressure chamber in the HPP processor into a thermally isolated vessel. Hence, adiabatic conditions should appear during compressing and later decompressing after the stationary phase. It means that temperature changes occurring during processing are not dissipated to the environment through the metal walls of the pressure chamber. Figure 9 shows the ‘adiabatic unit’ designed and built for the ‘preliminary feasibility test’ experiment.

The unit was placed within the HPP pressure chamber, the crucial element of the semi-lab scale HPP processor presented in Appendix A. Water was used as the medium transmitting pressure (outside the unit), and the medium inside the unit is shown in Figure 9.Only the adiabatic heating associated with compressing was tested in the experiment’s first stage.In the second stage of the experiment, the ‘barocaloric element’, shown in Figure 8, was placed inside the unit presented in Figure 9. Subsequently, temperature changes on compressing were scanned.

Regarding, the first stage, the issue of the adiabatic heating caused by compressing is often recalled in reports and reviews on HPP and HPT processing with the statement that for water 100 MPa (1 kbar), compressing causes 3–4 K temperature rise. The results presented in Figure 10 show that the pattern of temperature change depends on the onset reference temperature: it is ‘concave’ for $T_{ref.}=5\;{}^{\circ}C$, ‘convex’ for $T_{ref.}=60\;{}^{\circ}C$, and linear for $T_{ref.}=40\;{}^{\circ}C$. For the first two cases, linear approximation for low pressures (in red) and just below P=600 MPa (in blue) strongly differs, as indicated in Figure 10. For the reference temperature $T_{ref.}=5\;{}^{\circ}C$ compressing up to P=600 MPa yields the temperature change $\Delta T_{adiab.}\approx 12.9\;{}^{\circ}C$, for $T_{ref.}=40\;{}^{\circ}C$ one obtains ${\Delta T}_{adiab.}\approx 21\;{}^{\circ}C$, and for $T_{ref.}=60\;{}^{\circ}C$ such compressing yields ${\Delta T}_{adiab.}\approx 24.5\;{}^{\circ}C$. These results and supplementary studies extended up to 95 °C allowed obtaining the empirical relation that can support studies and applications exploring the barocaloric effect of water:(16)$$\Delta T_{adiab.}\left(T,\;P=600\;MPa\right)=11.8+0.271T-0.0011T^{2}\;\;\;\left({}^{\circ}C\right)$$

Fundamentally, it is related to the Clapeyron Equation (4), but its application requires knowledge of the pressure changes in specific heat for each isotherm.

In the experiment’s second stage, the barocaloric element presented in Figure 8, filled with neopentyl glycol (NPG), was used. NPG was the first material in which

For such an experiment, the preliminary knowledge of the pressure dependence of the orientationally disordered crystal (ODIC)—solid crystal discontinuous transition temperature $T_{D}$ is required. Figure 11 presents the results of studies yielding such data. They have been obtained by monitoring dielectric constant changes on compressing, for which a sharp change in detected values appears when passing the transition. It is shown in the recent report of the authors related to dielectric studies in neopentyl glycol [11]. This report and ref. [9] present essential components of the given experiment. The Plastic Crystal—Crystal phase transition’s pressure changes coincide with the results reported in ref. [48]. However, the high resolution of results presented in Figure 11 enabled the scaling of obtained results. It is shown by the blue curve in Figure 11, described by the Simon–Glatzel equation originally derived for the pressure dependence of the melting temperature [111,112].
(17)$$T_{D}\left(P\right)=T_{ref}{\left(1+\frac{P}{\Pi }\right)}^{1/b}$$
where: $T_{ref}=T_{D}\left(P=0.1\;MPa\right)=307.9\;K$, Π=431 MPa, b=5.65.

The high-temperature phase is related to the orientationally disordered crystal (ODIC) phase, where molecules are translationally frozen in a crystalline network, but for molecules creating this network, the orientational freedom remains. For the crystal phase, both the translational and orientational freedoms are frozen. The basic reference for the barocaloric is the Clapeyron equation, generally developed for the solid–liquid transition; see Equation (4). Originally ΔH is the enthalpy (‘latent heat’) associated with the transition; for the $liquid\to solid\left(crystal\right)$ transition reflects the internal energy excess of the liquid (disordered) phase, dissipated to the surrounding, when fusing into the solid phase. The absorption of the energy from the surroundings, related to $\Delta H^{\prime }$ is needed for liquid←solid. When the process occurs when compressing or decompressing under adiabatic conditions and in a space-limited container, there is a rise or decrease in temperature. It is the manifestation of the barocaloric effect.

So far, the barocaloric effect has been considered the most promising concept for the new generation of thermal energy storage, focusing on ‘cool’ storage, which is released upon decompressing and can be applied for supporting refrigerators or air conditioners [47,48,107,108,109].

However, the final application still requires overcoming two significant challenges: (i) the method of effective exchange of thermal energy between the interior of the pressure chamber and the environment; the current designs of pressure chambers show the level of difficulty (see Appendix A); and (ii) hysteresis, i.e., the difference between the fusion (solidification) and melting pressures, matched with a slight difference between in enthalpy changes.

These limitations are unimportant for the barocaloric effect-supported HPT innovative device discussed in the report. Moreover, it explores the barocaloric effect on compressing, leading to the surrounding heating, and decompressing, leading to the surrounding cooling. For the ODIC←→Crystal discontinuous transition, the orientationally disordered ODIC phase can be considered as the parallel of the disordered liquid phase in the above discussion regarding the Clapeyron equation. The ODIC mesophase in NPG locates this material in the family of plastic crystal-forming materials, which is worth stressing. When considering mechanical properties, plastic crystals resemble plasticine and are strongly susceptible to pressure. Therefore, plastic crystal-forming materials are located within the soft matter category. For the solid crystal phase, orientational and translational degrees of freedom are frozen, and it is mechanically ‘hard’, i.e., by decades less susceptible to pressure than the ODIC phase. This difference in compressibility, or susceptibility to pressure, described here is one of the essential causes of the colossal barocaloric effect in NPG.

In the second stage of the experiment, the element shown in Figure 8 was filled in with m=10 g of NPG and placed in the ‘adiabatic unit’ shown in Figure 9. The initial reference temperature before compressing was established at 60 °C, i.e., ~25 K for which the discontinuous phase transition occurs at $P_{D}\approx 300\;MPa$. It is related to the entropy change [48] ΔS~450 J/Kkg. Following this, one can expect ~30 K temperature rise due to the barocaloric effect in the given case, taking also into account the mass of NPG and the mass of water in the adiabatic element shown in Figure 11. The total temperature rise, linking the adiabatic heating and the barocaloric effect contributions, should not exceed ~362 K (~90 °C) to avoid exceeding the $T_{D}(P)$ curve, which would lead to a discontinuous phase transition from the solid crystal to the ODIC phase under P=600 MPa, leading to energy absorption and then cooling. It is notable that for compressing up to 800 MPa, heating up to ~110 °C is possible. The results of this experiment are shown in Figure 12.

The above results show the feasibility of the innovative ‘HPT-barocaloric’ concept presented in the preceding section. From the perspective of applications, locating a significant part of the barocaloric material in the walls of the cylindrical container where the product is placed can be convenient. Elements shown in Figure 8, dispersed among products, can support action, particularly the speed and uniformity of the heat energy distribution. Using other barocaloric-effect-yielding materials can increase the efficiency and flexibility of the barocaloric process. For instance, for liquid-solid crystal transition in higher alkanes, the super colossal values ΔS=700–800 J/Kkg were reported (2022, [109]). It reduces by 50% the amount of required barocaloric material in comparison with NPG. The ‘steeper’ pressure dependence of discontinuous transition temperature $T_{m}\left(P\right)$ than in NPG opens the possibility of the barocaloric effect-related temperature rise ~60 K, already at P~400 MPa. Even higher ΔS values can be expected for the liquid crystal—solid crystal phase transition in liquid crystal-based nanocolloids [113].

The results presented in the given report show the feasibility of the supported innovative barocaloric effect. So far, the significant motivator of extensive barocaloric effect research is the future application in thermal energy storage facilities. Such innovation is very important in the energy crisis and sustainable civilization times. However, for applications, challenging problems requiring breakthrough solutions still exist. As discussed above, such barriers are absent for barocaloric effect-supported HPT technology.

### 3.3. High Pressure Processing: Socio-Economic Aspects and New Challenges

The Industrial Revolution epoch essentially changed human civilization, influencing the rise of the global population, as indicated in the Introduction [6]. Its success can be linked to overcoming successive grand crises through massive implementations of subsequent innovative technologies. However, it would not have been possible without the conceptual base of the Scientific Method and feedback interactions with emerging socioeconomic innovative solutions. The latter is the topic of subsequent sections.

#### 3.3.1. Comments on HPP Market Development

This sub-section focuses on developing HPP and HPT for foods and their global-scale market. It presents the functional scaling relation describing changes in industrial scale processors, enabling reliable forecasting of future evolution, which can parallel the total HPP market development.

Massive implementations of innovative technologies such as thermal pasteurization, freezing and cooling, or various product-focused chemical preservatives solved one of the grand problems of the Industrial Revolution epoch. These innovations were introduced at the end of the first Industrial Revolution and remain dominant. In the 20th century, they were up to date with Ohmic or microwave heating [114,115], which can act quickly on the total treated volume.

In recent decades, however, it has become clear that the breakthrough successes associated with new-generation technologies for food and beverage preservation and processing also have a dark side related to the initiation of growing pandemics of several ‘civilization’ diseases or undesirable environmental impacts.

As indicated above, innovative high-pressure technologies such as HPP (‘cold pasteurization’) or HPT (‘relatively cool’ sterilization) can be considered another technological response to these new challenges, this time for 4th and 5th Industrial Revolution times. For HPP, the ‘global market experiment’ has already achieved significant success.

Figure 13 shows the changes in the number of industrial HPP processors from 1990 to 2022, based on data from refs. [116,117,118,119,120,121,122,123,124]. The central part of the figure is a standard bar plot for such presentations. In the inset, the same data is presented in the semi-log scale, revealing the simple pattern of functional changes. The plot reveals, that two decades ago (~2004), an exponential (‘geometric’) increase in the number of HPP industrial processors appeared, which continues to this day:(18)$$N\left(\Delta Y\right)=6.5\times exp\left(0.15\times \Delta Y\right)$$
where *N* denotes the number of processors ΔY=Y−1989.

Graphically, such changes are proven by the linear relationship shown in the inset. An interesting consequence of such a relationship is that the number of HPP processors will more than triple in the next 6 years: $N\left(2030\right)\to 3700$. The current global value of the HPP food market is estimated at USD 7.3 billion and is expected to double after 2032 [78]. 

The growing importance of HPP technology for ‘other’ products such as cosmetics, pharmaceuticals, or pet foods is worth noting. Currently, it can be estimated at ~20%. Figure 14 shows current estimates of global HPP technology applications for various food products.

#### 3.3.2. New Innovative Products, New Markets

The Industrial Revolution epoch essentially changed human civilization, influencing even the pattern of the global population rise—as indicated in the Introduction. One of the fundamental reasons for global success was solving successive grand challenges by mass implementing innovative technologies. However, all these would not have been possible without the conceptual support of the Scientific Method and feedback interactions with emerging socioeconomic innovative solutions.

So far, the primary goal of HPP applications is to ensure microbiological safety at the level of pasteurization, i.e., 5-log decay of pathogenic microorganisms.

However, other beneficial effects of high pressure have been used for some products. An example can be facilitating the mussel’s separation from the surrounding shell. This critical feature for gourmets results from the very high susceptibility to compression of the mussel muscle and the practical lack of compressibility for the solid shell.

For the authors, the latter property also opens up the possibility of broader use of this difference in compressibility, i.e., the change in volume due to the action of pressure. For a food product, it can reach $V\left(600\;MPa\right)\sim 20\%$, while for a solid-state component of food, e.g., shell, crust, or bone $V\left(600\;MPa\right)\sim 0\%$.

The authors want to indicate yet another innovative possibility associated with using HPP processors: the ‘cold’ pressure sterilization process for biofilm removal. Its spontaneous appearance is a huge problem in medicine and the pharmaceutical and food industries.

Nowadays, biofilm removal is possible via mechanical cleaning, and in some cases, it is supported by aggressive and, therefore, environmentally hazardous liquids. Medical equipment must be supported by long-term heating at 160–250 °C [125]. The problem is particularly challenging for elements containing polymer pieces or surfaces that are difficult to access. The difference in compressibility of the biofilm formed on the surface of a solid-state element, and the same solid-state element itself leads to the removal of the biofilm on compressing. The authors carried out preliminary tests confirming experiments for steel plates and PVDF (polyvinylidene difluoride) at a pressure of P=600 MPa for 10 min at T=30 °C. Significant was the supplementary impact of the rapid decompression terminating the treatment. Tests used the HPP processor, as shown in Appendix A. Plastic and steel plates were placed in a bag made from a polymer film usually used for HPP foods filled with distilled water.

The above results show that exploring the unique possibilities of differences in high pressures’ influence on ‘soft’ and ‘hard’ materials can be important. Other examples of possible new implementations can be indicated: (i)When HPP technology is used for raw meats, there is a problem for beef or pork related to reducing their characteristic reddish color. It is not the desired view of the product by the consumer. However, it results from the process occurring evenly in the entire volume. The degree of activation of the process depends on the value of the maximum pressure, the time of its application, and the temperature. Therefore, the product can be ‘pressurized’ (‘pre-cooked’) to a different extent defined by the parameters (P, t, T). It is a possibility of qualitative extension of the ‘low-temperature cooking’ technology. Here, additional opportunities are opened by applying the innovative HPT technology described in the previous section.(ii)A few years ago, a patent was issued for the possibility of quickly and evenly soaking various food products in an appropriately prepared marinade, but also, for example, beetroot juice, orange juice, … or even appropriately diluted chocolate. It can be supplemented by the ‘cold precooking’, solely by compressing. The patents [126,127], in which one of the authors of this work participates, show an application for vegetables and, above all, meats. The effective action has already been obtained for P~200 MPa for a few minutes at room temperature.(iii)Extraordinary pro-health features of green parsley-based nectars are widely known [128], but rapid biodegradation and separation into two layers were major problems.

A few years ago, the patent of the author of this work et al. [129] was related to ‘deep’ chopping of green parsley to a micrometric level with the addition of water and for flavoring a small number of chopped fruits (kiwi, orange, apple, etc.) to obtain nectar. The ‘designed’ HPP treatment led to microbiological safety, up to 3 months, and the strongly limited tendency to the separation into layers.

HPP technology to ensure the microbiological safety of fruit or vegetable juices is currently one of the primary and most widely used applications. However, in the case of multi-component fruit juices, the option of changing the dominant taste appears, with different HPP application scenarios regarding the values of parameters (P,T,t) as shown by the authors’ experience [129].

(iv)Another known option is the possibility of pressure ‘cold’ cooking of eggs, to a varying extent depending on the parameters (P,T,t) of the HPP process.

Figure 15 shows a photo of eggs treated this way, with a taste different from the classic thermal preparation, depending on pressure, temperature, and time of HPP treatment.

(v)One of the features of food products after HPP technology is often a velvety structure and taste, which has already been noted for the first market products using this technology in 1990/1991. The structure of many products, such as jams or hummus, is wholly homogenized, related to the new taste values mentioned above [130].(vi)For ground coffee, coffee beans or coarsely granulated coffee beans placed in water at a temperature of 6 °C to 50 °C (HPP) or even up to 120 °C (HPT) led to the creation of a new generation of beverages with a variety of flavors with different pressure-time implementation scenarios, as has been tested by the authors of the given report. It can create new generations of beverages using cocoa, tea, and herbs placed in ‘cold’ water and compressed using an HPP processor.(vii)For cow’s or goat’s milk or cream, the action of pressure in the range of 300–400 MPa allows for achieving microbiological safety but also permanently influences the size of micelles, which determine many values of milk, including taste [131,132].

The above examples show that HPP and HPT technologies can be a source of qualitatively new food products that offer new functional and taste values while maintaining unique health-promoting properties. It means that there is a chance for qualitatively new pro-health food products. There are also new options for gourmets and innovative chefs, opening up unique creative possibilities for dishes with extraordinary tastes and flavors. For such implementations, HPP processors with volume V=5–10 L, seem optimal. Such devices are already available at ~EUR 200,000–300,000 [85].

Perhaps the most extraordinary feature of HPP and HPT technologies is the ability to reconcile consumers’ expectations (high quality and freshness of the product) with the preferred conditions for producers and logistics. Significant is the durability extended up to 3–6 months, using only minimal cooling of 6–8 °C. HPP and HPT also naturally limit product losses and disposal costs. In this context, HPP and HPT technology implementation, with the appropriate support of modern logistics and AI-supported management, can revolutionize the market of pro-health, ‘targeted’ catering, primarily for hospitals and sanatoriums but also for seniors and convalescents at home or simply for people who want to change their diet to a genuinely pro-health. Thanks to the extraordinary features, it is possible to prepare high-quality meals in single locations, focusing on specific health needs of patients or customers. They can be immediately stored in automated cold stores (6–8 °C). Then, based on doctors’ or dieticians’ prescriptions/recommendations, focused HPP/HPT-treated dishes were delivered directly to patients and customers. The meals can be delivered in divided packages into parts requiring or not requiring pre-heating, for 2–3 min only.

It opens the route to precisely plan for diets supporting the pro-health goal for each person. The dream and wish of Hippocrates of Kos, the symbolic father of medicine, can become a reality ‘*let food be thy medicine and medicine be thy food*’ [133]. It was only a *memento* for 2400 years, but now it can be introduced globally. A proposal for such a system from the producer to the patient and client is shown in Figure 16.

The specific features of HPP and HPT technologies discussed above offer still other business opportunities, namely:The elements of the system presented in Figure 16 could be used in some branches of the catering business, for example, in popular eateries selling meals ‘by weight’. The significant problem of such restaurants is associated with losses that occur with unpredictable changes in the number of customers or the utilization of remaining foods, resulting in significant losses and costs. The aforementioned use of products based on HPP technology, ready to eat even after 3 min, solves this problem.HPP and HPT technologies open up qualitatively new possibilities for creative gourmet chefs. The implementation of high pressures opens up a new dimension of food preparation. It could be an innovative option for haute cuisine restaurants. For such applications, HPP processors with a pressure chamber volume V=5–10 L could be sufficient. Such a volume means a reasonable price and simple operations and services. It is worth emphasizing the possibility of extending such a processor to the HPT option using the barocaloric effect-based innovation presented above.There is a wide range of local, traditional products and dishes with excellent quality, pro-health, and taste values. Usually, they are unavailable outside their traditional regions. They are based on locally available products with exceptional qualities. Local implementation of HPP and HPT technologies for these products and dishes could open broad markets, benefiting local communities and producers greatly. This can also be a qualitatively new offer for a global customer.

#### 3.3.3. HPP Market and Consumers’ Economic Constraints

A combination of favorable economic conditions, social needs, and expectations can strongly stimulate the market development of innovative technologies. However, factors limiting this development can also appear. These issues for HPP and HPT technology implementations are the subject of this section.

A significant socio-economic aspect of HPP and HPT technology development is the expansion to geographical regions where it is currently absent or weakly represented. Currently, over ~67% of the market is in North America, namely the USA, Canada, and Mexico, with populations of 333.3 million, 39 million, and 127.5 million, respectively. Together, this is a population of 500 million people, with the USA having the dominant market share. The current population of Europe is estimated at 750 million, including the European Union ~450 million, and countries outside the union, such as the United Kingdom, with a population of 69 million. This wealthy part of the world currently accounts for only ~18% of the HPP market. In the European Union, this technology’s natural area of expansion is ‘new countries’ that have been linked since 2004. Their share and importance in the European Union economy have increased significantly. For Poland, the largest country in this group, the gross domestic product has increased in the last two decades from ~255 billion USD in 2024 to 865 billion in 2024 and is expected to exceed 1000 billion in 2028 [134,135]. Figure 17 shows changes in wages during this period for the highest-earning employees in industrial enterprises associated with more than 100 people [134,135]. It can be representative even for ~40% of people working in Poland. A systematic income increase is visible. Interestingly, the empowered exponential relation can portray it well:(19)$$S\left(t\right)=S_{0}exp\left[{\left(\frac{\Delta t}{\tau }\right)}^{\beta }\right],\;\Delta t=t-t_{ref.}$$
for $t_{ref.}=2008$, $S_{0}=S\left(t_{ref.}\right)=3800\;PLN$, τ≈18.2 (year).

The red curve in the figure shows the behavior described by Equation (14). Values of obtaining parameters suggest the 50% salary rise in 2024 value (8600 PLN~2000 EUR) occurs for ~12 years, i.e., in 2036. The inset shows the results of the derivative-based analysis recalling Equations (7) and (8). Particularly noteworthy is the strong upward trend after 2016, the first traces dating back to 2012. It is worth mentioning here a model discussion of the importance of the exponent β in empowered exponential behavior, where β>1 indicates the emergence of internal trends strengthening the internal energy of the process.

The authors of this paper are associated with the Laboratory, which conducts research and development work related to HPP technology, including preliminary works for entrepreneurs interested in implementing HPP technology for their food products. Since 2014, entrepreneurs have been growing interested in such work, which correlates well with the changes in income shown in Figure 13. It may mean that in a relatively large country such as Poland, a growing group of consumers has emerged, ready to take an interest in more expensive, innovative, and health-promoting food products. The first still-imported HPP products have begun to appear on store shelves. Small service centers for this technology have also emerged, with significant support from state funds. The growing interest in HPP technology was evident from the discussions between the authors of this paper and entrepreneurs. Everything was changed by the economic consequences of the outbreak of the global COVID-19 epidemic and, to an even greater extent, by Russia’s attack on Ukraine (2022) and the still ongoing war. Poland is particularly affected by the energy crisis, the million-strong influx of refugees, and the almost front-line expansion and armament of the army necessary for the state. At present (August 2024), the economic situation of Poland seems to be stabilized on a national scale. Inflation is ~2.7%, unemployment is 4.7%, and in many regions of the country below 3%, and the expected GDP growth in 2024 is 2.7%. These are very favorable parameters on a strategic scale. However, the personal economic situation is difficult on the scale of a significant part of society, especially from the large public sphere and services where incomes are lower than results from Figure 16. This is primarily due to the huge increase in electricity prices, and, to a dominant extent, gas and coal are necessary in the Polish climate for heating for at least 6 months a year. For example, gas prices in the period mentioned above increased by more than 4× (!). This is also reflected in other fixed costs, such as rents and taxes related to housing. It means that the real purchasing power of the population in Poland has hardly changed since 2020, and for some quite large social groups, it has fallen significantly, especially in 2023. In practice, this also means a large increase in the debt of many citizens, related not to investments or plans but solely to the payment of current bills. The authors of this work do not believe that economic circumstances support the market implementation of HPP products, which are necessarily premium products. This conclusion is confirmed by the practical disappearance of products of this type from store shelves, except for specific stores in Warsaw (the capital) and several comparable urban centers. Apart from these centers, there is also a visible change in the range of food products towards cheaper, ‘economic’ products and promotional brands. It is worth noting here that Poland is not only a representative country in the EU but also over-representative due to those mentioned above formal, excellent general characteristics of economic development. In the case of Poland, there is still a substantial economic and psychological burden due to the war in neighboring Ukraine. According to the authors of this work, due to unfavorable global economic changes that occurred after 2020, where the irreversible consequences of the great energy crisis are of particular importance, in Poland and probably other ‘new’ EU countries, the market area for more expensive food products, including those related to HPP technology, has been qualitatively reduced. An important factor is probably also the greater pessimism related to the energy and climate crisis and the extremely difficult geopolitical situation. However, there may be significant niches for possible implementations of HPP technology where the beneficial pro-health features of the processed product will be associated with lower prices for the consumer. Such a potential possibility may be offered by the new directions of the use of HPP technology mentioned in this work.

The analysis of the socio-economic situation in Poland shows that despite its multitude of beneficial features, the HPP market development can be hindered—at least for the basic path focused on microbiological safety. The emergence of a large consumer group is important for HPP and HPT market success. In Poland, housing costs range from 48% of the average salary (median) in Warsaw to 30% in the province. In the latter case, energy costs can exceed another 30%. In the Polish climate, they are primarily related to dramatically rising electricity or gas prices for heating, even for 7–8 months a year.

The second factor is consumer optimism about the future. Despite its relative wealth in Europe, there is a significant problem with these factors. On a global scale, there are still other huge and populated areas for implementing HPP and HPT technologies, with a tiny share in this area. These are Asia (~8%), South America (~3%), and Africa (~1%). The key here seems to be the end of the current geopolitical and war turmoil and the re-entry of the world economy onto a wisely managed pro-development path.

## 4. Conclusions

High pressure preservation/processing (HPP) and high-pressure and high-temperature sterilization are scientific and technological responses to the challenges of the fourth and fifth Industrial Revolution epochs related to the challenges of the 21st century times of a sustainable society, where not only the quantity of food is significant but also its pro-health feature or qualitative reduction of adverse impacts on the environment. High-pressure preservation/processing (HPP) and high pressure and high temperature (HPT, HPHT) sterilization are scientific and technological responses to the challenges of the fourth and fifth Industrial Revolution epochs related to the challenges of the 21st century times of a sustainable society, where not only the quantity of food is substantial but also its pro-health feature or qualitative reduction of adverse impacts on the environment. These methods, especially HPP, have already achieved great market success and huge research interest. In 2023 alone, the phrase ‘HPP, high pressure, food’ refers to a huge number of 4890 works. Over the last three years, at least 30 review publications have been published on the detailed implementation of the mentioned technologies for various specified types of food.

This publication refers to the mentioned review works as necessary and valuable but does not repeat the factual evidence therein. This publication is a ‘progressive resume’, recalling the issues related to HPP and HTP technologies to the extent necessary for their definition and understanding but focusing on essential topics rarely mentioned or requiring direction. On this basis, they further discuss the development of these technologies that may open up new and perhaps groundbreaking options for them. First of all, these are:The innovative HPHT (HPT) technology is based on the barocaloric effect, which allows for action at high temperatures only when maximal stationary pressure acts on the product. The high-pressure value controls the triggering and termination of the high-temperature simultaneous operation stage.Proposal for using HPP processors as a base for cold sterilization, particularly for removing biofilms from fools or other equipment important for medicine, the pharmaceutical industry, and the food industry. It can be essential for elements built from metal and ‘plastic’ elements.

Notable are also:There is a new discussion regarding the post-HPP decay of microorganisms, particularly the introduction of the innovative distortion-sensitive analysis.The indication of the feedback interactions between research, technological, and socio-economic issues is a characteristic feature of the Industrial Revolution’s epoch.

The latter point is particularly worth stressing. The last two centuries have seen a remarkable development in feedback interactions between technology and the underlying fundamental science base, with the methodological background of the Scientific Method. This report first recalls the influence of the first Industrial Revolution epoch on the development of still-dominated ‘classic’ food preservation methods. New generation methods such as HPT and HPT are discussed as possible solutions for the great problems of ‘classic’ methods, which are poorly acceptable in the fifth Industrial Revolution and Sustainable Society epoch.

## Figures and Tables

**Figure 1 foods-13-03028-f001:**
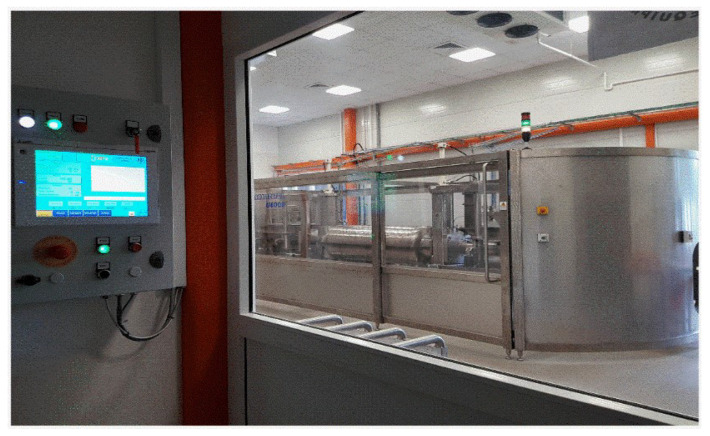
The HPP processor with a high pressure chamber volume V=50 L, an automatic product loading and unloading system, and external control via a visible panel. The processor operates as part of the HPP pilot line in the X-PressMatter Lab for pressurized soft matter, foods, and glasses in IHPP PAS Innovation in Celestynów near Warsaw. The construction was completed by the Unipress Equipment unit of IHPP PAS (Poland). The pressure chamber, safety covers, and the control panel in a ‘control’ room are visible.

**Figure 2 foods-13-03028-f002:**
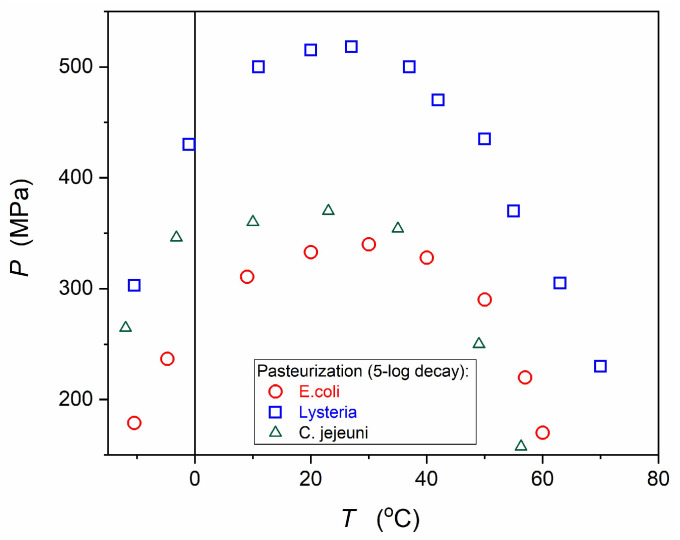
Pasteurization curves associated with 10^−5^ (5-log) decay of selected bacteria, given in the plot, for tests in the Agar matrix in the pressure-temperature (P–T) plane. The tests were carried out in the X-PressMatter Lab IHPP PAS.

**Figure 3 foods-13-03028-f003:**
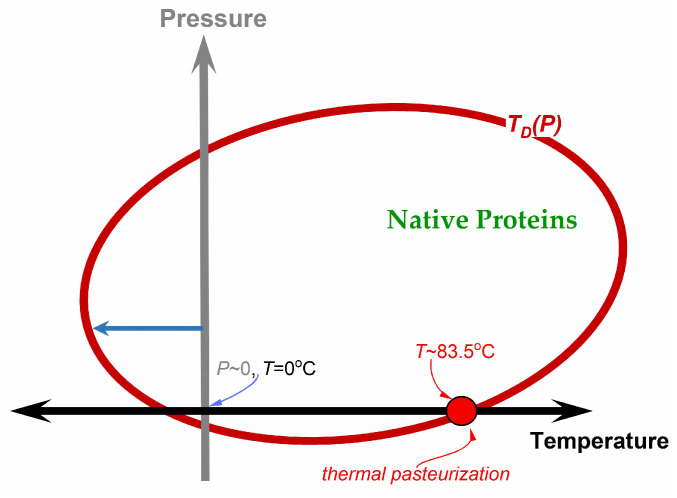
The scheme of the denaturation curve in P–T plane. It was prepared following the Claussius–Clapeyron equation model analysis by Smeller and Herremans [91,92]. The location of the ‘classic’ thermal pasteurization is indicated.

**Figure 4 foods-13-03028-f004:**
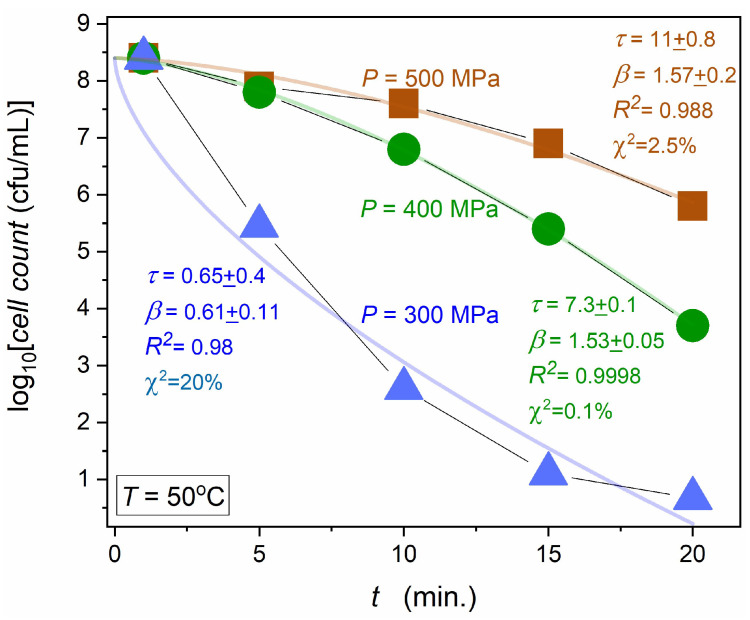
Survival of *S. aureus* (ATCC 6538) in human milk treated with high pressure pulses with ta maximal ‘stationary’ values 300 MPa, 400 MPa, and 500 MPa, at the temperature 50 °C, lasting t=5, 10, 15, and 20 min. The process temperature is close to the standard Holder protocol conditions used for the thermal ‘soft pasteurization’ at T=65 °C for 30 min. Experimental data directly recalls results from ref. [69], and the parameterization is related to the empowered exponential Equation (8). Values of parameters, with errors, are given in the plot.

**Figure 5 foods-13-03028-f005:**
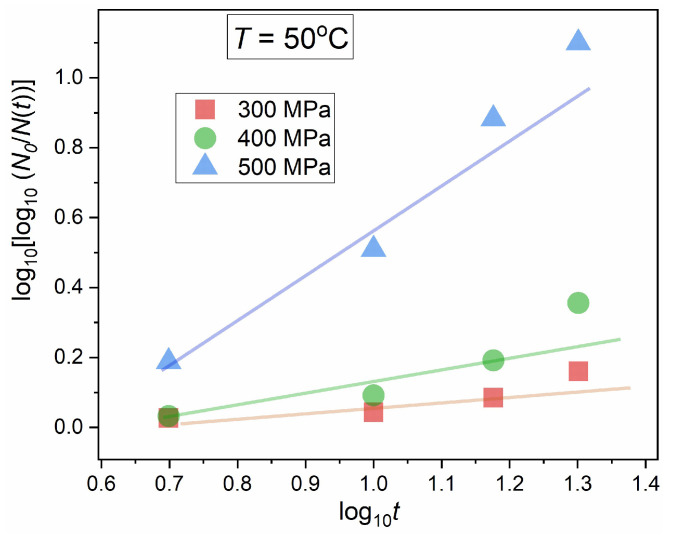
The linearized plot related to Equation (11) focused on the empowered exponential Equation (9). It is related to experimental data given in Figure 4 for the decay of *S. Aureus* in human milk after HPP tests at 300 MPa, 400 MPa, and 500 MPa.

**Figure 6 foods-13-03028-f006:**
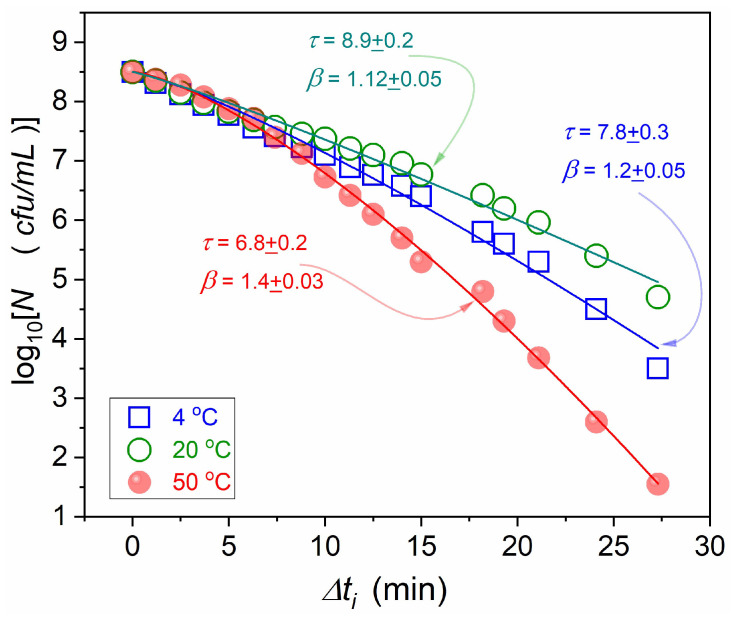
The decay of *Staphyllococcus aureus* amount in human milk after different compressing times for three isotherms noted in the plot, using the double pulse HPP+ pattern [81]. It is the semi-log scale plot for which the simple exponential decay is manifested by linear behavior. Curves portraying experimental data are related to the empowered exponential decay given by Equation (9).

**Figure 7 foods-13-03028-f007:**
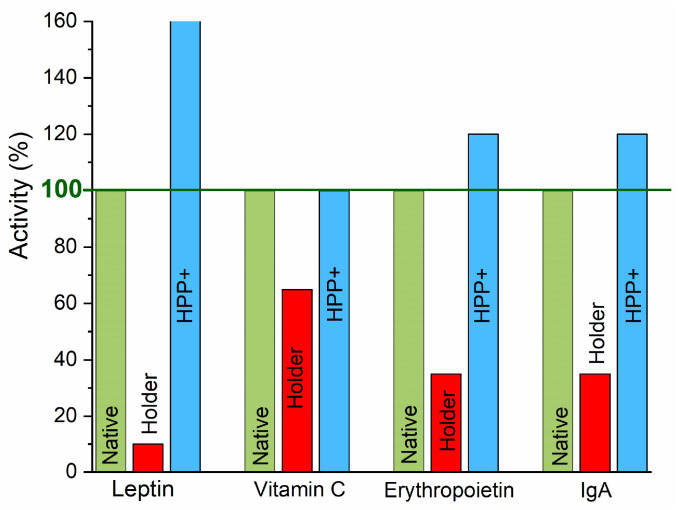
The activity of selected crucial constituents of human milk after the standard ‘soft, Holder, pasteurization’, and the high-pressure processing (HPP+), compared to the native milk. However, the HPP for food technology has some significant limitations, particularly visible when recalling numerous excellent HPP features beneficial for consumers (fresh product quality or pro-health features) and for producers and logistics, such as the extended shelf-life or higher prices for better quality (based on results from refs. [81,82]). The results are for HPP+ technology defined by two pressure pulses with stationary values P=200 MPa and P=400 MPa, each lasting 15 min, and a break of 30 min between. Processing temperature T=20 °C. As shown in refs. [81,82], it allows microbiological safety comparable to processing with a single pulse at the stationary value P=500 MPa and the temperature T=50 °C, but without negative impacts on essential nutritional and immunological properties, which for human milk appear already at a pressure of P=500 MPa.

**Figure 8 foods-13-03028-f008:**
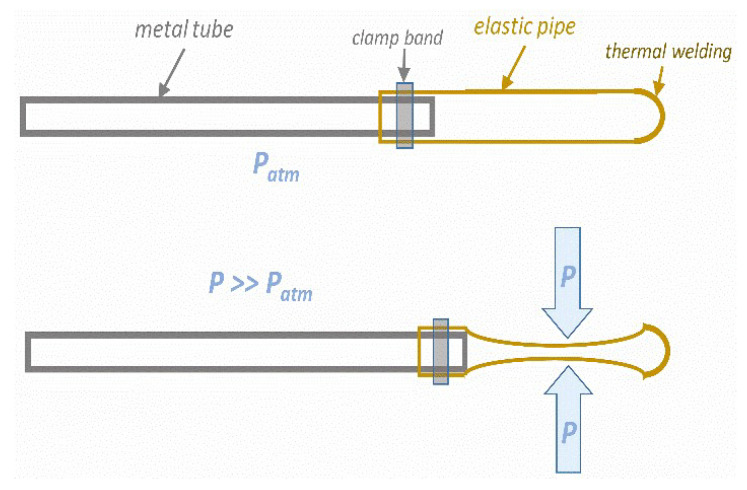
The ‘element’ scheme for the HPT is supported by the barocaloric effect. The barocaloric effect-related material is placed inside the element, and pressure is transmitted via the deformation of the elastic tube. The action of pressure is indicated.

**Figure 9 foods-13-03028-f009:**
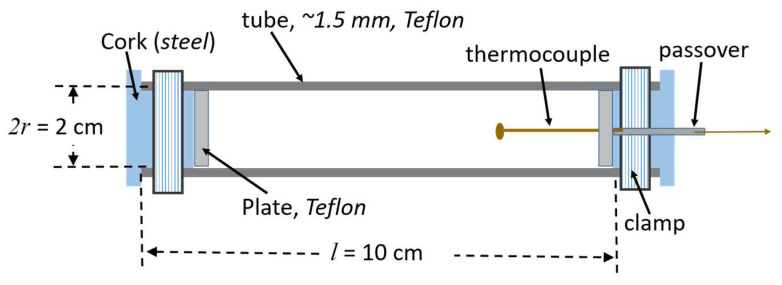
The element for obtaining adiabatic condition under compression after placing inside the pressure chamber: in the given case related to the HPP processor presented in Appendix A, Figure A2. All elements are described, and their sizes are scaled to given dimensions. Pressure is transmitted via the deformation of the tube’s wall on compressing. In the experiment, a copper-constantan thermocouple was used. It was linked to a microvoltmeter, an enabled temperature scan with ±0.02 K resolution.

**Figure 10 foods-13-03028-f010:**
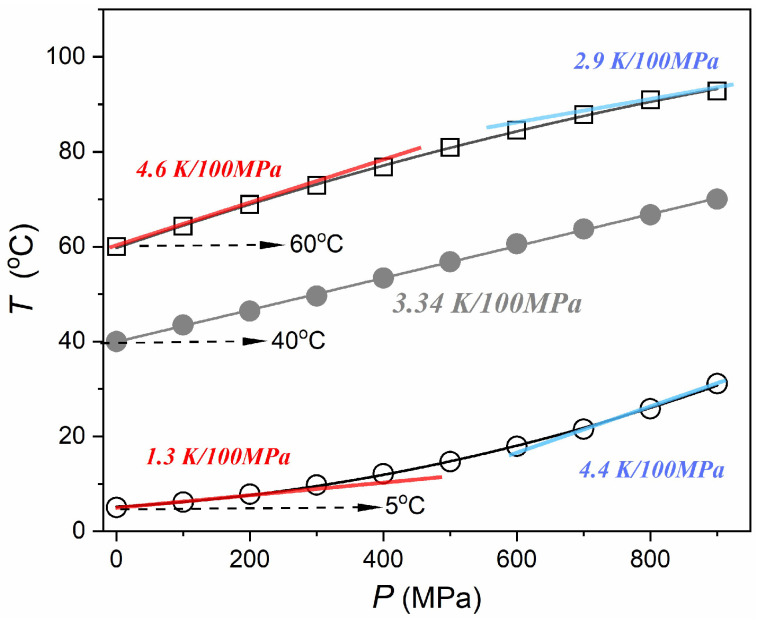
Changes in water temperature upon adiabatic compression, starting from the reference temperature indicated by dashed arrows. The results have been obtained using the experimental unit presented in Figure 9.

**Figure 11 foods-13-03028-f011:**
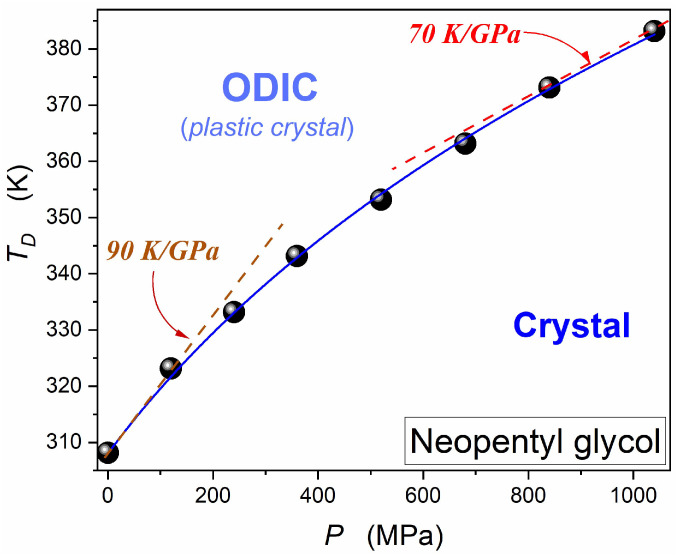
Changes of the discontinuous phase transition temperature in neopentyl glycol, determined by dielectric constant scans on compressing for subsequent isotherms.

**Figure 12 foods-13-03028-f012:**
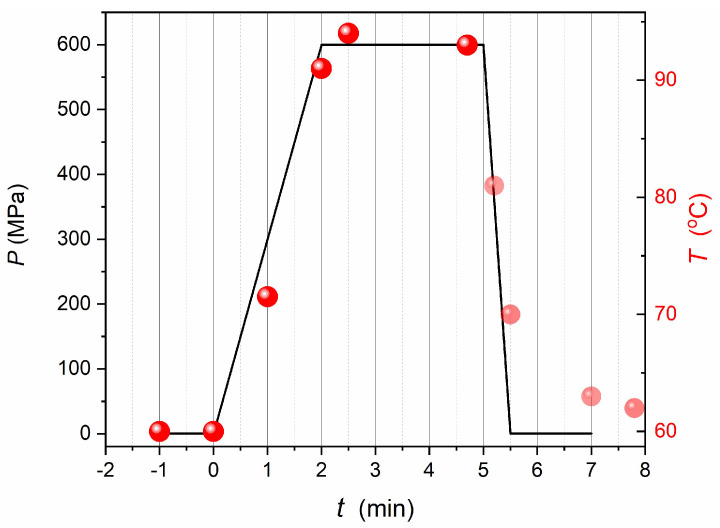
Temperature changes inside the unit shown in Figure 8 on compressing up to P=600 MPa. The left scale shows the time-related profile of the pressure pulse, and the right scale is related to detected temperature values. The right scale and related data points are in red. For decompressing, temperature data are shown in light red. The temperature scale (left) is adjusted to the pressure scale (right) to facilitate the view.

**Figure 13 foods-13-03028-f013:**
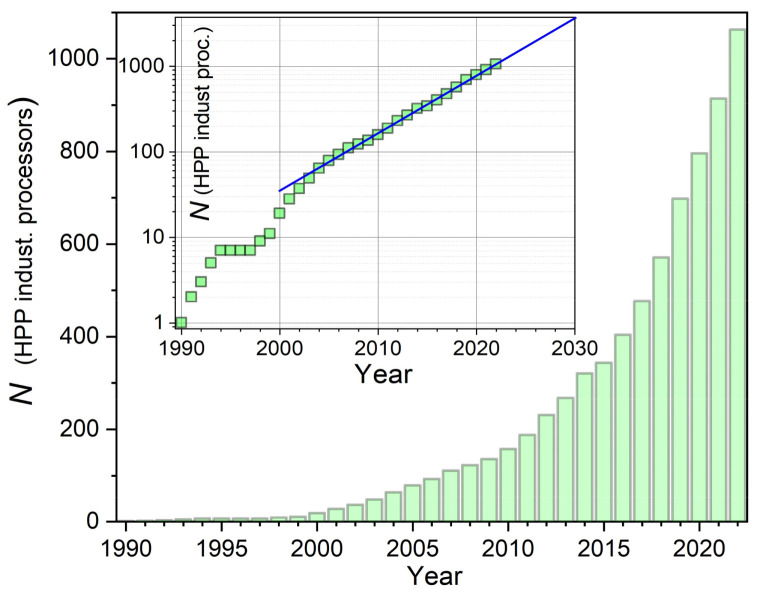
Changes in the number of HPP processors with the high-pressure volume chamber V>100 L. The inset shows the semi-log plot of data from the central, standard’, bar plot, revealing the exponential behavior (Equation (15)) proved by the blue line. The plot has been prepared using data from refs. [116,117,118,119,120,121,122,123,124].

**Figure 14 foods-13-03028-f014:**
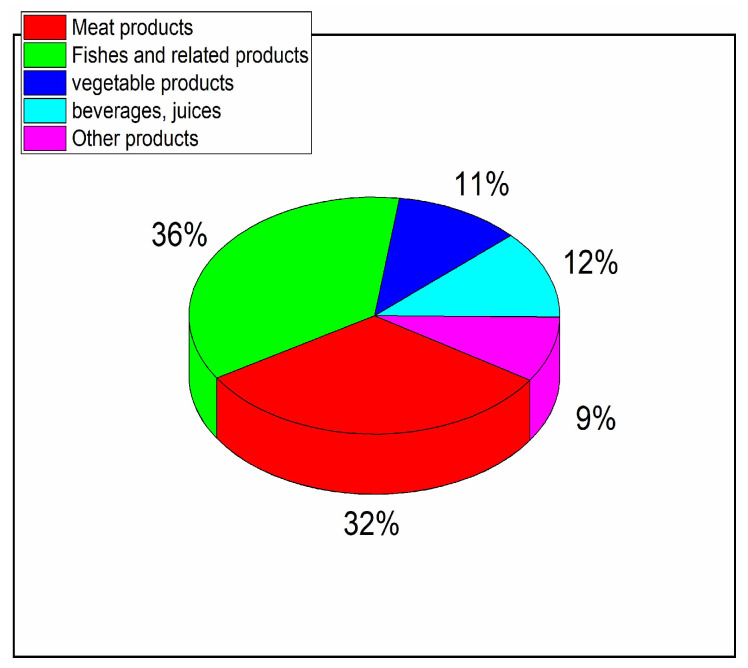
Estimated distribution of HPP technology implementation for food on a global scale nowadays. Estimations have been prepared via the authors’ analysis of data from refs. [116,117,118,119,120,121,122,123,124].

**Figure 15 foods-13-03028-f015:**
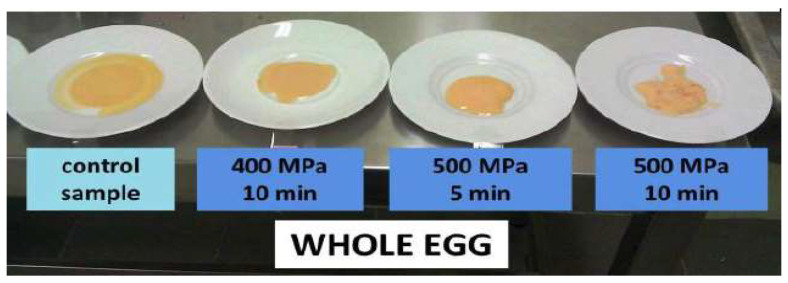
Egg: white and yolk after treatment with different pressures at different times in the HPP process.

**Figure 16 foods-13-03028-f016:**
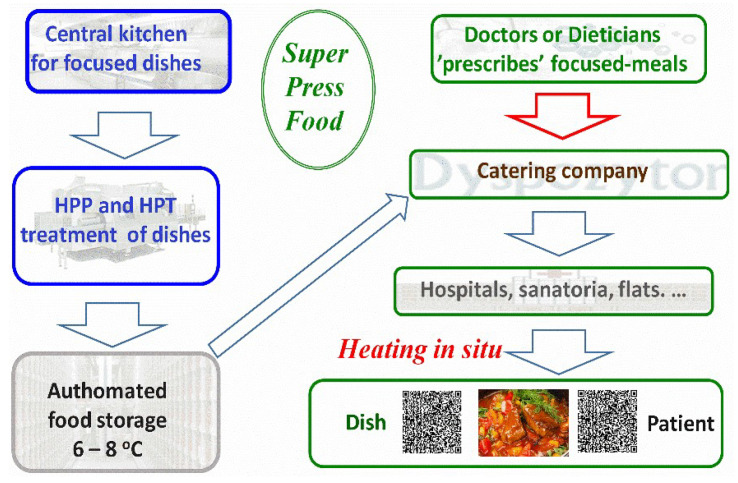
The general distribution scheme of HPP and HPT-treated dishes focused on specific pro-health requirements.

**Figure 17 foods-13-03028-f017:**
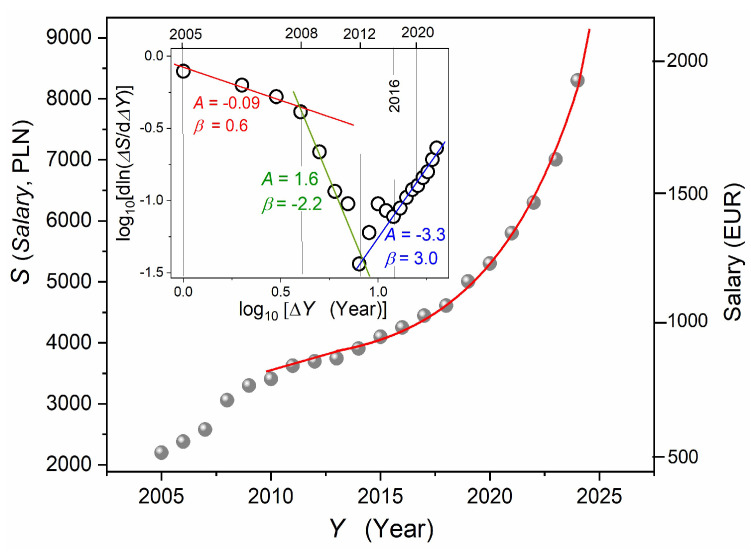
The central part is the average gross salary (S) in the enterprise sector in Poland from entering UE. For the currency, 1 EUR=4.3 PLN was assumed as the most representative in the presented period, particularly in the last decade. The inset shows the distortion-sensitive analysis related to Equation (13) and the portrayal via empowered exponential Equation (19). Values of the power exponent and the parameter A in Equation (16) are given: ΔY=Y−2004 (year) and ΔS=S(Y)−S(2004) (PLN). The reference year 2004 is related to Poland’s accession to the European Union, based on data from refs. [134,135].

## Data Availability

The original contributions presented in the study are included in the article, further inquiries can be directed to the corresponding author.

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
