# Peer review of "Food Preservation in the Industrial Revolution Epoch: Innovative High Pressure Processing (HPP, HPT) for the 21st-Century Sustainable Society"

_foods, 2024, doi:10.3390/foods13193028_

Round 1

Reviewer 1 Report

Comments and Suggestions for Authors

See the comment in the attached file.

Author Response

REVIEWER#1

General comments

  1. Reviewer#1: ‘The review identifies problems with commonly used nonlinear fitting models for predicting the decay of microorganisms in HPP but does not elaborate on the nature of these problems or provide a comparative analysis of alternative models.’  

Response: It is difficult to agree with this remark. I understand, however, that it results from an attempt to avoid criticism of previous results, because a clear presentation of a given issue leads directly to it. It is included in the new text presented between LINES 449- 474.

  1. Reviewer#1: ‘The review should include a detailed critique of existing models, perhaps with case studies or specific examples where these models fail. It should also provide a rationale for the proposed changes in modeling paradigms and o?er empirical validation through experiments.

Response:   the response to the remark (1). above is also significant to the given point. – Hence, see the new text between LINES 449 – 474, which explains the meaning of further results presented in this section

Particularly, the meaning of the further part of this subsection is shown in depth, and the empowered exponential and the ‘double exponential’ models are discussed.

Note: the discussion includes new issues – not discussed so far(i) the new meaning of the fitted parameter, (ii) the innovative derivative-based and distortions sensitive analysis; (iii) it is also maybe the first discussion regarding the reliable fitting error and fitting quality within HPP studies topic.

  1. Reviewer#1: ‘While the review mentions that it doesn’t repeat specific characterizations of HPP and HPT as available in other review papers, this might leave readers without the necessary context to fully understand the challenges and innovations discussed.

Response:   The repeated statement is mainly related to extensive tables,  usually presented in review reports in which results of HPP (mainly) and HPT applications with respect to different pathogenic microorganisms and the impact of such treatment are presented. The latter is most often heuristic. Such extensive presentations are based on ‘literature scans’. This fact is now stressed at the end of the Introduction. Other significant features regarding relevant issues of HPP and HPT are in the paper. The report also presents significant issues that have not been presented so far. It was one of the targets of this paper. See added explanation: Lines 166-173.

  1. Reviewer#1: ‘A concise summary of key characterizations relevant to the challenges being discussed should be included. This will ensure the review is self-contained and accessible to a broader audience.

Response: Following the suggestion each section/subsection of the report now starts from (added) short,  ‘summaries of key characterizations relevant to the challenges being discussed.’ I am grateful for this suggestions, it seems that it really makes the report ‘easy to follow’.

  1. Reviewer#1: ‘The introduction of the barocaloric e?ect for HPT is innovative, but the review seems to lack experimental or theoretical evidence supporting its feasibility. Without data, this concept remains speculative.

Response: Now, such a preliminary feasibility test is presented in the new section 3.2.3.  ‘Preliminary feasibility test for the ‘HPT-barocaloric’ innovative solutions

  1. Reviewer#1: ‘Include preliminary experimental data or simulations to demonstrate the barocaloric effect's potential eff. Additionally, the review should address any practical challenges in implementing this concept, such as equipment limitations or scalability.

Response: First, we want to note that two of the authors have carried out R&D works on the barocaloric effect for a few years. The barocaloric effect is considered the most promising concept for the next generation of ‘cool storage’ for use in refrigerators and air conditioners. Nevertheless, there are still no pilot implementations of such technology due to the so-called hysteresis and the problem of transporting hot/cold from the chamber and to the chamber. These factors are completely irrelevant to the proposed innovative HPT barocarics-based solution. This fact and technological simplicity may cause it to be the first practical use of the barocaloric effect.

This explanation has been added at the end of section 3.2.2. ; see the new section 3.2.3.  ‘Preliminary feasibility test for the ‘HPT-barocaloric’ innovative solutions

  1. Reviewer#1… While the review highlights the importance of reliable microbial decay modeling, it would benefit from a more in-depth discussion of the specific limitations of current models and the potential advantages of the proposed distortion-sensitive routine.’

Response:   This issues was also indicated by other Reviewers. Hence, please see the revised section 3.1.3.   paticularly see the new text between Lines 451-473.

  1. Reviewer#1…’ The review appears to discuss significant technological advancements and challenges but lacks empirical data to support these claims. For instance, the issues with nonlinear fitting models and the barocaloric e?ect concept are mentioned without providing data to back these assertions.

Response:   For nonlinear fitting routines used for decay of microcorgansism the above statement is not true. Please the discussion related to  Figs. 4, 5, and 6.  They present the fisrt-ever such type analysis (!)

This issue was addressed in the 1st version of the manuscript and now , following the Reviewers suggestions, it  is  ‘clarified & addressed’ in the revised manusript, Section  3.1.3.

For the barocaloric effect, this problem has already been discussed above. The results are presented in the new section 3.2.3.  ‘Preliminary feasibility test for the ‘HPT-barocaloric’ innovative solutions response this problem. We want to indicate the really grand experimental difficulty of obtaining such evidence. It was possible only due to the extraordinary possibilities of X-PressMatter Lab IHPP PAS,  included into RoadMap for Significant Research Infrastructure plans, see the page: young4softmatter.pl

  1. Reviewer#1 Include more detailed experimental results or case studies to support the discussion. If the data is unavailable, the review should explicitly state this and outline a plan for future research to address these gaps.

Response:   All data are available on request. According to the rules for public institutions, they have to be placed in the year of the report acceptance in the public, open REPOD database, and for access, the doi of the report is only needed. Please note that some of the results and analyses are well beyond the current state of the art. See the analysis shown in Figs. 4, 5, 6. For Figs 6, the set of experimental data is also well above standards, which is clearly explained in the text. The result is presented in the the new section 3.2.3.  ‘Preliminary feasibility test for the ‘HPT-barocaloric’ innovative solutions is the first message on new, important finding and we plan soon publish the complete report only on this topic.

At the end of Conclusions, the emerging key quests – cognitive gaps are indicated as advised in the comment. Lines 1352-1372

  1. Reviewer#1The review touches on the feedback interactions between socio-economic and technological issues, but the depth of analysis in this area seems limited.

Response:    Such a general discussion is needed but it is beyond the scope of the given report, and rather a topic for the research project.

The target of this report was to indicate the significance of the mentioned feedback for advanced food preservation technologies, particulartly HPP – and in our opinion it has been reached.

  1. Reviewer#1 ‘Provide more quantitative analysis or case studies that explore the socio-economic impacts of HPP and HPT technologies. This could include market analysis, cost-benefit assessments, or surveys of industry stakeholders.’

Response:    This comment can be considered as the topic of the BIG and quite expensive research Project. It needs focused market and stakeholder research, advanced statistical analysis, etc.… Also, market reports from professional agencies would be needed. Only the commercial access prices 7 000 – 12 000 EUR for a single report. All these are well beyond the given project's possibilities and the target – which is the preliminary indicating the problem, hardly addressed so far.

Nevertheless, we added new comments regarding the HPP related situations in Poland. They are related to the Authors experience  - we are working in X-PressMatter Lab IHPP PAS (Poland) which carry out HPP food related research but also works for industry partners interested in the topic . See the Appendix.

  1. Reviewer#1A more detailed explanation of how the barocaloric effect can be harnessed for HPT

Response:    these issue has been explained above. Nevertheless, please see the added  section 3.2.3 regarding the ‘Feasibility tests’  and the supplemented reference list.

  1. Reviewer#1  
    • Given the emphasis on socio-economic factors, a more quantitative analysis of the economic constraints and potential business opportunities associated with HP and HPT would provide valuable insights.

13.2   The review discusses economic constraints and emerging business opportunities but may not provide enough data or analysis to substantiate these claims.

13.3    ‘The conclusions drawn about economic constraints and business possibilities should be backed by robust economic models, market data, or detailed case studies. The review should critically evaluate these aspects, possibly by consulting with industry experts or referencing market research reviews.

Response:    These points indicate a good program for a BIG and EXPENSIVE Projects, and cannot be addressed in the given report. This report shows only the preliminary and probably the first indication that the review discussion on HPP and HPT for foods requires a more ‘holistic’ approach linking science, technologu and socioeconomy. These three directions and their feedback interactions are in the focus of the Industrial Revolution epoch.

Nevertheless the meaning of such approach is strengthen and indicated in Conclusions as the cognitive gap requiring further studies. See the end of Conclusions, where thi issue is recalled.

  1. Reviewer#1The summary does not mention whether the research was funded by industry stakeholders who may benefit from positive findings, nor does it discuss any potential biases in the presentation of data.

Response:    The only funder (or rather supporter)– National Center for Science (Poland) is mentioned. It is a project related to energy storage, including the barocalorics.’ See the added statement in lines 1377, 1378.

  1. Reviewer#1The review should include a clear declaration of any funding sources, afiliations, or potential conflicts of interest. Transparency in these areas is crucial for maintaining scientific integrity.’

Response:   This issue was explicitly explained above. See lines lines 1376-1378.

  1. Reviewer#1  A)  ‘There is no mention of ethical considerations, such as the environmental impact of new technologies or the safety implications for consumers.

B ) The review should address any ethical concerns related to the implementation of HPP and HPT technologies. This might include discussions on sustainability, food safety, and the potential impact on small-scale food producers.

Response:  These problems directly result from all presented aspects. This issue is recalled numerous times when the practical lack of waste for HPP and HPT processing is stressed. The lesser than for classic thermal processing energy requirements are also strongly stressed.  It is also indicated in the new ‘Feasibility’ section that the innovative HPT solution supported by the barocaloric effect requires essentially less energy than any solution developed so far.

Note – that in this report – for the first time – the problem of packing disposals is indicated. All these are strictly associated with Ethics and Sustainability.

We are also grateful for indications for further research, which have been taken into account in the revised Conclusion.

Yours Sincerely Prof. dr hab. Sylwester J Rzoska

IHPP PAS (IWC PAN ‘Unipress’)

Warsaw, Poland

Reviewer 2 Report

Comments and Suggestions for Authors

In this manuscript, the authors reviewed HHP and HPT applied on foods. They indicated significant problems with commonly used nonlinear fitting model routines, showing the significance of changes in the applied paradigm and the final validation via the distortions-sensitive routine. Moreover, the authors provided an innovative concept based on the barocaloric effect for HPT technology. The manuscript would be valuable for readers concerning high pressure, as well as food processing. The manuscript would be published after minor revising according to the reviewer’s comments and suggestions as follows:

Figure 1: The authors should provide a schematic figure of the apparatus with the picture for readers’ convenience.

Figure 2: In the legend box, E-Coli should be E. coli.

Figure 2: The reference of data used for the figure should be added. If the data were provided by the authors, description should be added in the text.

Figure 5: The label of Y-axis should be corrected. According to Eq. (10), “N(t)/N0” should be “N0/N(t)”?

Figure 7: In the text, there is no description about Figure 7.

Figure 8 and text lines 740-: The authors are encouraged to add more detail working mechanism for the “proposed element” for readers’ convenience.

Figure 10: The authors should add more information on this figure. Present of future? Reference should be added.

Line 799: 3.2.2 should be 3.3.2.

Comments on the Quality of English Language

 Quality of English Language would be sufficient, but the reviewer suggests minor editing of English language.

Author Response

REVIEWER #2

‘…..The manuscript would be valuable for readers concerning high pressure, as well as food processing. The manuscript would be published after minor revising…’

  1. Reviewer#2:Figure 1: The authors should provide a schematic figure of the apparatus with the picture for readers’ convenience.’

Response:  the system required for HPP studies, in details, is presented in the Appendix. It is shown in a way allowing a reader to see the challenge of such studies.

  1. Reviewer#2:Figure 2: In the legend box, E-Coli should be E. coli.

Response:   it has been done.

  1. Reviewer#2:    ‘Figure 2: The reference of data used for the figure should be added. If the data were provided by the authors, description should be added in the text.

Response: it has been done.

  1. Reviewer#2:Figure 5: The label of Y-axis should be corrected. According to Eq. (10), “N(t)/N0” should be “N0/N(t)”?

Response:   It has been corrected.

  1. Reviewer#2:Figure 7: In the text, there is no description about Figure 7.

Response:   the correct recalling regarding Figure 7 and 6, have been done. See lines 552-559 and line 622 for Fig. 7.

  1. Reviewer#2:Figure 8 and text lines 740-: The authors are encouraged to add more detail working mechanism for the “proposed element” for readers’ convenience.’

Response:  It has been done. See extensive explanations between lines 839-866.

  1. Reviewer#2:Figure 10: The authors should add more information on this figure. Present of future? Reference should be added.’

Response:  It has been done. See the supplemented caption.  Note. that the new Section 3.2.3.‘The feasibility test’, introduced  to follow comments of Rev.#1,  required 4 new Figures, and then shifted a number of figures below, including the former Fig.10

  1. . Reviewer#2:Line 799: 3.2.2 should be 3.3.2.’

Response:  Done

Reviewer 3 Report

Comments and Suggestions for Authors

The paper is two long. The introduction is vague and does not focus well on the problem or the challenges of technology. The inclusion of the results of the experiments carried out by the authors on human milk do not provide relevant information for commercial food processing. I suggest to remove this part from the manuscript.

No summarizing table of the main results related to the technology has been provided. 

Minor comments: 

Lines 249-242: The main phase is keeping the product under the constant planned high pressure, lasting from 3𝑚𝑖𝑛 to 10𝑚𝑖𝑛. Then, the final phase begins, namely decompressing, removing tens/hundreds of liters of water from the chamber, opening the chamber, and shifting the product away to the product feeder. This phase also requires ~5𝑚𝑖𝑛.

In my opinion the times are shorter 1 second-10 min and decompression is almost instantaneous in commercial units.

Figure 2: Explain the source of the data/results

Figure 3: Explanation of this figure in the manuscript is not clear.

Line 383: parasitic microorganisms is not adequeate, should not it be pathogen microorganisms? 

Lines 386-390: Human milk should not be a model for all food types to analyze the effect of HHP. It is not a commercial product.

Figure 4. Staphylococcus aureus is the most pressure resistant microorganims present in human milk? Should it serve as model to decide pressure conditions to obtain a safe product?

Figure 5. Why 50ºC was chosen? is this initial temperature? At 500MPa and initial temperature of 50ºC, the real temperature would be even higher than that required for HoP (if compressiong heating is taken into account)

Line 531. Figure 5 or Figure 7?

Figure 7: Detail HHP processing conditions (pressure and holding time)

Figure 10 Add data source

Author Response

REVIEWER#3

The reviewer was a bit critical of the report, namely: ‘The paper is too long. The introduction is vague and does not focus well on the problem or the challenges of technology. The inclusion of the results of the experiments carried out by the authors on human milk does not provide relevant information for commercial food processing. I suggest to remove this part from the manuscript. No summarizing table of the main results related to the technology has been provided. ‘

Response to the above ‘general issues’ :

  • Reviewer #3:The paper is two long. The introduction is vague and does not focus well on the problem or the challenges of technology.

Response

  • the length of the report is OK for Reviewers #2 and #4. For Review#1 it should be extended by other results/issues (suggested by the Reviewer), i.e., it should be longer for Reviewer#4.
  • The introduction shows the significance of technological innovations matched with socio-economic innovations as crucial factor for the success for the Industrial Revolutions and Sustainable Society epoch. This approach address the ‘heart’ of this epoch and exactly reflects the title of the report. It is illustrated for food preservation methods.

In my opinion, this is the first report with such interdisciplinary addressing the problem.

  • Reviewer #3: ‘The inclusion of the results of the experiments carried out by the authors on human milk do not provide relevant information for commercial food processing. I suggest to remove this part’.
  • Response: this comment does not agree with other Reviewers. Human milk is the life most important food. For the authors the most import was the availability of ‘own’ extreme quality experimental data – unfortunately very hardly available for ‘classic’ foods .

The presented analysis explicitly shows that many similar studies on ‘classic food’ have essential reliability problems. This is the Novelty and Significance of these results. See the conclusion at the end of Section 3.1.3.

  • Reviewer#3:No summarizing table of the main results related to the technology has been provided. ‘
  • Response: The is a huge number of reports on HPP for foods, and the vast majority presents such Tables. Only in 2024 (till August): at least 15 (!), and almost each of these reports presents ‘HPP implementations Table’ based on extensive analysis of literature.

The target of the given Report was not to repeat these facts, but to supplement earlier review report by issues not addressed so far. They are socio-economic sustainability problems, the new ‘microbiological’ analysis (based on the milk example) or the HPT innovation as the future challenge proposal.

 ‘Minor corrections’ indicated by Reviewer #3

  1. Reviewer#3:Lines 249-242: The main phase is keeping the product under the constant planned high pressure, lasting from 3??? to 10???. Then, the final phase begins, namely decompressing, removing tens/hundreds of liters of water from the chamber, opening the chamber, and shifting the product away to the product feeder. This phase also requires ~5???.’   -    In my opinion, the times are shorter 1 second-10 min, and decompression is almost instantaneous in a commercial unit

Response:  I have a long practice in the realistic HPP implementations, and I cannot agree with the 1 second time scale suggested by the Reviewer. Unfortunately, manufacturers of industrial HPP processor often present non-realistic values to suggest more cycles per hour, i.e., better – but not real – efficiency.

It is not possible to decrease the first filling stage – this is limited by the pump efficiency and the huge volume.  For the last phase, decompressing in/a few seconds is formally possible, but such ‘shock action’ in large volume created a so-called ‘pressure hammer’, which is destructive for many elements of the pressure chamber and processors. I have extreme experience in high pressure R&D and my advice ‘Don’t do it’. Manufacturers can advise extremely short decompression, but expensive problems start just when the guarantee passes. Hence, honestly, I can only change the text as follows:

CURRENT:  ‘ This phase also requires ~5???. ‘      to

REVISED: ‘This phase can last from ~ 1 second for the sudden decompressing to  ~5??? for the soft decompressing’. Lines 246-248.

  1. Reviewer#3:Figure 2: Explain the source of the data/results’

Response:  it has been done (see the caption)

  1. Reviewer#3:Figure 3: Explanation of this figure in the manuscript is not clear.’

Response: the extended explanation is given in Lines 368-376. It required ‘splitting’ Eq. (4) to Eqs. (4a) and (4b) It also includes recalling the most recent report by Laszlo Smeller [2022, IJMS] on this topic.

  1. Reviewer#3: ‘Line 383: parasitic microorganisms is not adequeate, should not it be pathogen microorganisms? 

Response:   it has been corrected.

  1. Reviewer#3 ’ Lines 386-390: Human milk should not be a model for all food types to analyze the effect of HHP. It is not a commercial product.

Response:  Yes, human milk is not a commercial product. But commercial model good does not exist. One can only consider a set of subjectively selected commercial food products. However, this is beyond the scope of the given report. This issue is well presented in tens or (even) hundreds of review reports on HPP.

Human milk is the most important food and the only common food for any human on Earth. It is also a ‘product’ of extraordinary complexity and a multitude of actions and impacts. See Lines 417-423.

  1. Reviewer#3 ‘Figure 4. Staphylococcus aureus is the most pressure resistant microorganims present in human milk? Should it serve as model to decide pressure conditions to obtain a safe product?

Response: Staphylococcus aureus is the essential pathogenic microorganism for human milk, and in the given case, it is the main focus of any studies.  See comment in Lins 422,423.  

  1. Reviewer#3 ‘Figure 5. Why 50ºC was chosen? is this initial temperature? At 500MPa and initial temperature of 50ºC, the real temperature would be even higher than that required for HoP (if compressing heating is taken into account)’

Response: “The standard thermal preservation technology for human milk is called Holder pasteurization, known as ‘soft pasteurization’. It is processing at 65oC for 30 minutes. Destructions in nutritional, bioactive, and immunological properties caused by the standard pasteurization at ~86oC -  even for a short time, are unacceptable for human milk. The temperature T=50oC, is the highest temperature considered in HPP tests on human milk, where the thermal pasteurization non-desired impacts can be avoided. See the extended caption to Fig. 7.

Regarding the increase in temperature caused by compressing, notable that for standard HPP processors, this issue is now extensively explained in the new section 3.2.3. See also Lines 701-707, 585-593.

  1. Reviewer#3:Line 531. Figure 5 or Figure 7?’

Response:  it has been corrected

  1. Reviewer#3:Figure 7: Detail HHP processing conditions (pressure and holding time’

Response:   Please see the extended caption to Fig. 7,

  1. Reviewer#3: ‘Figure 10 Add data source’

Response:   it has been added. 

Reviewer 4 Report

Comments and Suggestions for Authors

Foods Preservation in the Industrial Revolutions Epoch: Innovative High Pressure Processing (HPP, HPT)

for the 21st-Century Sustainable Society

 Abstract: Is provided and supported sufficient information.

 Introduction:

The manuscript should be reorganised. I suggest that all the formulas be pulled out from the Introduction and presented in a separate section.

Some of the informations provided in the sub-section should be placed in the introduction as part of the lietrature review. The manuscript is intended as a book chapter, but as it stands, the reader may become confused about whether the article is a review or a book chapter. Please revise the manuscript accordingly.

Overall, the work is interesting and has potential if it can be presented in a more fluid way. 

Comments on the Quality of English Language

Minor correction is required.

Author Response

REVIEWER#4

‘Overall, the work is interesting and has potential if it can be presented in a more fluid way’

Suggestions of Reviewer#4

  1. Reviewer#4 : ‘ Introduction: The manuscript should be reorganised. I suggest that all the formulas be pulled out from the Introduction and presented in a separate section.

Response:   In the introduction  only 3 basic equations, essential for the reasoning related to the ‘Sustainable Society & Food’ topic appears. Eqs. (2) and (3) are related to the basic Malthus model which introduces the link between population and food resources.

Eq. (1) shows that something very unusual happened 220 years ago and continues today. It is the Industrial Revolution epoch.

In my opinion, an informative and relatively simple mathematical equation can be more significant than many words. Hence, I do not see a possibility of separating these relations into a separate subsection.

Nevertheless, following the advice of the Reviewer, the Introduction has been tested, for the more fluent narration. Particularly, the meaning of the ‘multitude target’ of the given report, addressing issues weakly and not at all presented in available reviews, is stressed.

  1. Reviewer#4 ‘Some of the informations provided in the sub-section should be placed in the introduction as part of the lietrature review. The manuscript is intended as a book chapter, but as it stands, the reader may become confused about whether the article is a review or a book chapter. Please revise the manuscript accordingly.’

Response: The report has been tested following the above advice. Shifting some issues to the Introduction would not be in line with the suggestions of other Reviewers. Indeed, the report has been notably revised, also as required by Rev.#1, Rev.#2, and Rev.#3.

Please note that comments pf all , 4 Reviewers, re-shaped the report according to given suggestions

Round 2

Reviewer 1 Report

Comments and Suggestions for Authors

The authors significantly revised the manuscript, corrected all comments. The quality of the manuscript is good, and the contents are reasonable, therefore, it is suggested that the manuscript should be published in this journal.